

# Seismic signature of turbulence during the 2017 Oroville Dam spillway erosion crisis

Phillip J. Goodling[1], Vedran Lekic[1], Karen Prestegaard[1]

[1]Department of Geology, University of Maryland, College Park, 20742, USA

*Correspondence to*: Phillip J. Goodling (pjgood@terpmail.umd.edu)

**Abstract**

Knowing the location of large-scale turbulent eddies during catastrophic flooding events improves predictions of erosive scour. The erosion damage to the Oroville Dam flood control spillway in early 2017 is an example of the erosive power of turbulent flow. During this event, a defect in the simple concrete channel quickly eroded into a chasm 47 meters deep.

Erosion by turbulent flow is difficult to evaluate in real time, but near-channel seismic monitoring provides a tool to evaluate flow dynamics from a safe distance. Previous studies have had limited ability to identify source location or the type of surface wave (i.e. Love or Rayleigh wave) excited by different river processes. Here we use a single three-component seismometer method (Frequency-Dependent Polarization Analysis) to characterize the dominant seismic source location and seismic surface waves produced by the Oroville dam flood control spillway, using the abrupt change in spillway geometry as a natural

experiment. We find that the scaling exponent between seismic power and release discharge is greater following damage to the spillway, suggesting larger turbulent eddies excite more seismic energy. The mean azimuth in the 5-10 Hz frequency band was used to resolve the location of spillway damage. Observed polarization attributes deviate from those expected for a Rayleigh wave, though numerical modelling indicates these deviations are explained by propagation up the hillside topography. Our results suggest Frequency-Dependent Polarization Analysis is a promising approach for locating areas of

increased flow turbulence. This method could be applied to other erosion problems near engineered structures and to understanding energy dissipation, erosion, and channel morphology development in natural rivers, particularly at high discharges.



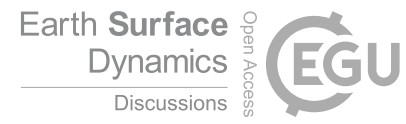

# 1 Introduction

Dam spillways are typically designed with features that generate controlled turbulent eddies, such as steps or changes in slope. These eddies entrain air into the flow, increase energy dissipation, and lower the mean flow velocity (Hunt and Kadavy, 2010a; Hunt and Kadavy, 2010b). Some of this dissipated energy is transferred as lift and drag forces on the bottom

of the spillway channel. If a defect in the spillway channel is present, increased turbulence and associated forces can quickly enlarge the defect, eroding the spillway and underlying embankment (USBR, 2014). In some cases, erosion propagates headwards, undermining the structural integrity of the dam (USBR, 2014). Structural elements and routine maintenance are designed to minimize these channel defects, however, they can develop quickly during extreme flows. Therefore, real-time monitoring of spillway turbulence during times of high release could provide early warning of the onset of erosion. Although

turbulence can be characterized with photographic images or measurements of velocity time series with submerged or overhead instrumentation, these procedures may be impractical on large structures or during catastrophic events. Seismic monitoring may provide a way to continuously evaluate turbulent intensity and associated erosion from safely outside channels or hydraulic structures.

Seismic waves have previously been used to characterize the geotechnical suitability of earthen dams and internal

dam seepage using passive seismic interferometry (e.g. Planès et al., 2016), but have not been used to characterize open-channel turbulence in dam spillways. Turbulence affects erosional processes in both hydraulic structures and natural rivers, therefore, techniques from the seismic river monitoring (fluvial-seismic) literature provide guidance. In the past decade, many authors have used near-channel seismometers to monitor rivers during monsoons (e.g. Burtin et al., 2008); natural floods (e.g. Govi, et al., 1993; Hsu, et al., 2011; Burtin et al., 2011; Roth et al., 2016) and controlled floods (Schmandt et al., 2013;

Schmandt et al., 2017). In many of these studies, the authors seek to separate the various sources of seismic energy, including precipitation, bedload transport, and flow turbulence (e.g. Roth et al., 2016). Bedload transport is traditionally difficult to monitor, therefore, research has been focused on isolating this source. Characterizing turbulence in rivers has been given less consideration in the fluvial-seismic literature, even though macroturbulent eddies place important controls on channel erosion (Franca and Brocchini, 2015) and may be important in spillway erosion. A forward mechanistic model by Gimbert, et al.

(2014) estimates the power spectral density of seismic energy produced by turbulently flowing water in a simple rectangular channel, in principle making it possible to use seismic data to invert for river depth and bed shear stress. This model, however, is based on assumptions of spatially uniform turbulence created by bed grain size; it ignores other sources of turbulence common in natural rivers and in engineered structures such as channel geometry variations. Recent work (Roth, et al., 2017) suggests that hysteresis between seismic power and discharge may also result from riverbed particle rearrangement, which

leads to different turbulent characteristics within the flow. This fluvial seismic body of work suggests seismic monitoring may be able to resolve changes in even in a dam spillway setting.

A near-spillway seismometer records seismic energy excited by a number of sources from different directions across a range of frequencies. These potential sources include primary and secondary microseisms, anthropogenic noise, wind, rain,

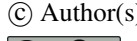



earthquakes, and nearby rivers. Without a way to differentiate among these sources by direction and frequency, interpreting seismic observations will be limited. This challenge was highlighted by Roth et al. (2016) and Roth et al. (2017), who indicated that the turbulent signal from a waterfall downstream of their study river reach may have dominated the observed low-frequency signals. Previous studies have attempted to locate the source of fluvial seismic energy by using arrays of

seismometers, primarily by observing the variability in seismic amplitudes around the river section of interest (Burtin et al., 2011, and Schmandt et al., 2017). A study by Burtin, et al. (2010) developed noise correlation function envelopes to identify segments of the Trisuli River that generated the most seismic energy at a given frequency. The greatest coherence between seismometer pairs (and inferred greatest seismic energy production) was located along river segments with the steepest river slopes and estimated incision rates. This approach is a promising one, though it requires an extensive array of seismometers.

A single-seismometer method for distinguishing various sources of seismic energy at different frequencies is more likely to be implemented in monitoring hydraulic structures and may be advantageous for fluvial seismic studies.

Discerning among seismic sources using a single station requires an evaluation of the three-dimensional ground motion recorded by a three-component seismometer. In traditional earthquake seismology, these motions indicate the arrival of body waves (P and S) and surface waves (Rayleigh and Love). For continuous ambient seismic sources such as turbulence,

the phase relationships between the signals in each component can provide information on the wave type and its propagation direction. Several researchers have suggested that turbulence may excite Rayleigh surface waves whereas sliding and rolling bedload transport may excite Love surface waves, though these authors relied on comparing the seismic power of the three components rather than analyzing phase relationships among the components (Schmandt et al. (2013); Barrière et al. (2015); Roth et al. (2015)). Recent forward models to estimate the power spectral density of seismic energy produced by moving

bedload and turbulently flowing water also assume that only Rayleigh waves are excited, though these assumptions have not been quantitatively tested (Tsai et al., 2012; Gimbert et al., 2014). Identifying the surface wave type excited by turbulent sources will help to identify the dominant mechanisms generating seismic waves in spillways and natural channels.

In this study we employ a single-seismometer method to observe variations in turbulence intensity and location within a dam spillway. Our goals are to 1) evaluate the scaling exponent between seismic power and discharge for different turbulence

and channel roughness conditions; 2) determine if a single-seismometer source location technique can be used to resolve changes in the location of flow turbulence in a spillway channel; and, 3) evaluate the surface wave type excited by spillway turbulence and erosion. The study site is the flood control spillway of the Oroville Dam, California, USA. Seismic and discharge data collected during the erosional event that damaged the flood control spillway in February and March 2017 provide a natural experiment for this study, during which a simple and straight channel was abruptly eroded into a complex

one.





## 2 Oroville dam crisis

The Oroville dam, located 100 km north of Sacramento, CA in the Sierra Nevada foothills, is the tallest dam in the United States (Fig. 1a). The dam spans the Feather River and provides hydroelectric power, flood control, and water storage for irrigation. Completed in 1968, the dam is constructed on Mesozoic volcanic rocks contained in the Smartville Complex

(Saucedo and Wagner, 1992). The dam is built adjacent to the Long Ravine Fault; therefore, a permanent seismic station was placed approximately 2 km from the dam site in 1963 to monitor possible reservoir-induced earthquakes (Lahr et al., 1976). Several studies have linked the unusually large drawdown and refilling of the reservoir in 1974-1975 to a 5.7 magnitude earthquake on 1 August 1975 located 12 km south of the reservoir (Beck, 1976; Lahr et al., 1976). In 1992, the Berkeley Seismological Laboratory installed a Streckeisen STS-1 broadband three-component seismometer at the site as station BK

ORV (BDSN, 2017). We are not aware of any studies that have investigated ground motion generated by the flood control spillway.

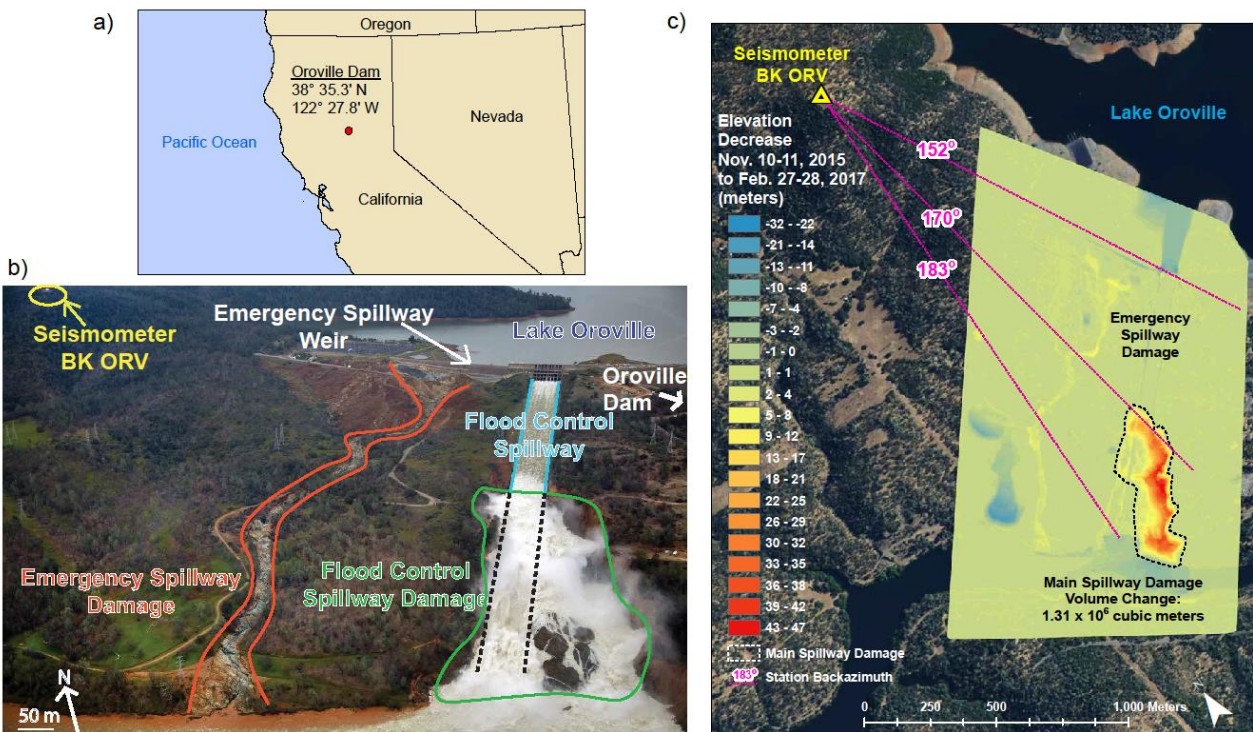

**Figure 1: a) Location of the Oroville Dam in Northern California. b) The damage created along the Flood Control and Emergency Spillways of Oroville Dam in February and March, 2017. The seismometer used in this study is located approximately 2 km from**
**the spillway. Photo credit: Dan Kolke, Department of Water Resources. Image taken on 2/15/2017. Estimated discharge during photograph is 2,800 m³ s⁻¹. c) A digital elevation model created from LiDAR points provided by the California Department of Water Resources. The elevation difference from a November 2015 elevation survey and a late February 2017 survey shows that the crisis incised a chasm up to 47 m deep. The volume of the main chasm is $1.3 \times 10^6$ m³. The incision resulting from the use of the emergency spillway is less than 20 m deep. The backazimuth (clockwise from north) in degrees is displayed for the top of the flood control**
**spillway, the top of the chasm, and the bottom of the flood control spillway. The seismometer is at an average 13° slope above the base of the flood control spillway and an average 8° slope above the top of the flood control spillway.**



At approximately 9 am PST on February 7th, 2017, during a controlled dam release of approximately 1400 m³ s⁻¹, a section of the concrete flood control spillway failed, leaving a defect in the spillway. A subsequent preliminary root cause analysis identified construction and maintenance flaws as the source of this initial defect (Bea, 2017; ODSIIFT, 2017a; ODSIIFT, 2017b). Ongoing heavy rainfall and runoff from the upstream watershed filled the reservoir to near capacity.

Reservoir managers increased the discharge through the damaged spillway in a series of tests and ultimately raised the discharge to over 1500 m³ s⁻¹. This discharge and associated high flow velocities resulted in turbulent scour around the defect, rapidly eroding the underlying embankment and incising a gully that bypassed the concrete spillway channel. Dam managers then limited the flood control spillway discharge to below 1800 m³ s⁻¹ (California Department of Water Resources, 2017a). High incoming discharge from the Feather River raised the reservoir level to capacity, which activated an emergency spillway

weir for the first time in the dam's 48-year history.

Discharges up to 360 m³ s⁻¹ flowed over the emergency spillway weir beginning at 8:00 am PST on February 11th while managers released approximately 1500 m³ s⁻¹ through the primary flood control spillway. Within 32 hours, rapid erosion at the base of the emergency spillway weir threatened to compromise its stability, triggering concerns of catastrophic failure. Managers increased the discharge through the previously damaged flood control spillway to 3000 m³ s⁻¹ and evacuated 180,000

people from the downstream city of Oroville, California. Elevated flood control spillway discharges lowered the reservoir level and stopped discharge through the emergency spillway weir on February 12th, 38 hours after activation. Elevated discharges continued through the damaged flood control spillway through the end of March, causing tens of meters of vertical incision into the weathered, sheared bedrock underlying the spillway (Bea, 2017). Figures 1b and 1c show the position of the seismometer and erosion incurred during the event. Using LiDAR data collected in 2015 and March 23rd, 2017, we compute

that 1.3 x 10⁶ m³ of material were removed from the flood control spillway damage area during the crisis, resulting in a vertical incision into the hillside of up to 47 m (Fig. 1c; see Supplemental Information) (California Department of Water Resources, 2017b).

## 3 Methods

### 3.1 Data collection and approach

In this study, we evaluate seismic signals detected during the Oroville Dam Erosion Crisis at broadband seismometer BK ORV, operated by the Berkeley Digital Seismological Network (BDSN, 2017). We divide the crisis period into five time intervals of constant discharge, each of which is longer than 15 hours in duration (Fig. 2). During each of these discharge intervals, channel geometry and discharge remain similar, allowing us to document the differences across intervals in the spillway-generated seismic signal. The five time intervals of interest are:

1) "Pre-Chasm" interval: 18 hours of ~1400 m³ s⁻¹ routine flood control spillway release before the initial spillway damage on February 7th,



2) "Emergency Discharge" interval: 38-hour interval when the emergency spillway weir was active and ~1500 m³ s⁻¹ was released through the flood control spillway

3) "High Discharge" interval: 78-hour interval when ~3,000 m³ s⁻¹ were released through the damaged flood control spillway,

4) "Post-Chasm" interval: 87-hour interval of ~1400 m³ s⁻¹ discharge through the damaged flood control spillway, and

5) "Zero Outflow" interval: 93-hour interval of zero discharge through the flood control spillway, which serves as a control interval.

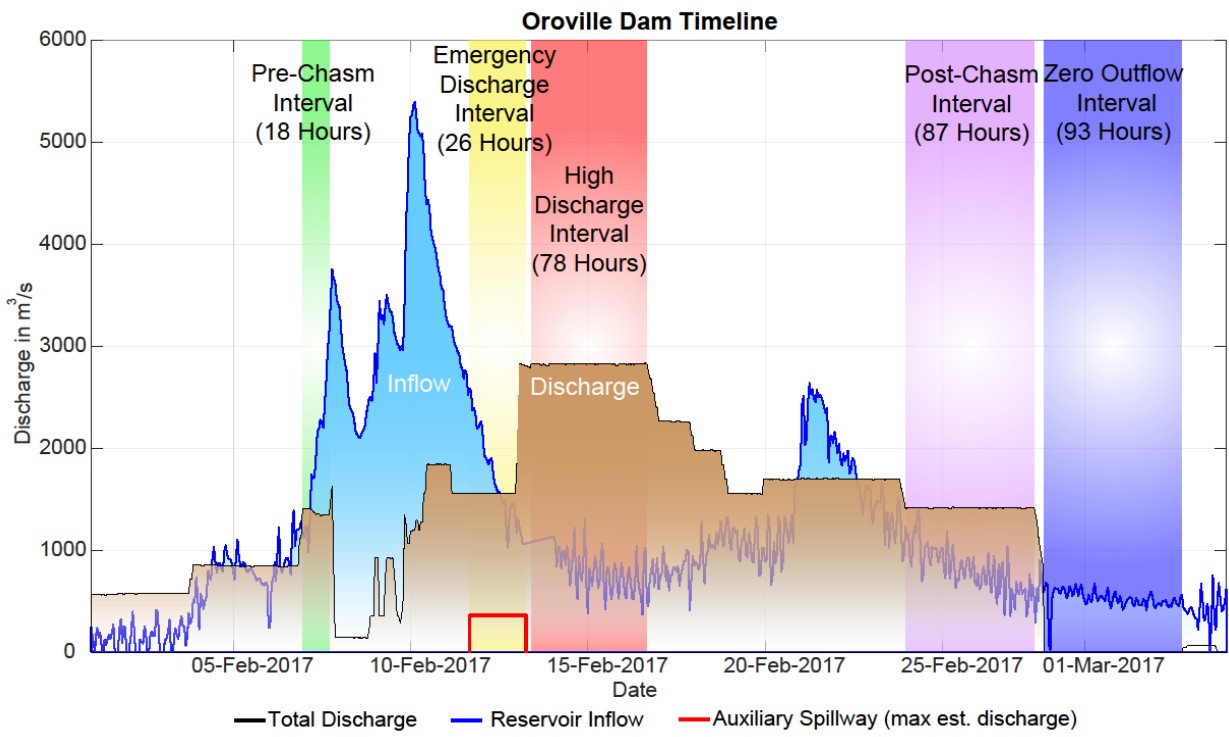

**Figure 2: Discharge and inflow at Oroville Dam in early 2017, as reported by the California Department of Water Resources. The five time intervals of constant discharge in early 2017 used in this study are highlighted and labeled. The "Pre-Chasm" and "Post-Chasm" time intervals have approximately equal discharge, but very different channel geometries. Data gaps in discharge and inflow data are linearly interpolated in this figure. The inflows reported are from the Feather River to Lake Oroville. The discharge displayed for the emergency spillway weir is the maximum reported by CA DWR media updates, as no quantified measurements have been published for this data.**

To encompass the erosion crisis period, we complied seismic data and spillway discharge data from 1/1/2017 to 4/1/2017. For comparison to the erosion crisis, we also compiled seismic data and spillway discharge for the second and third highest release periods during which continuous discharge and seismic data are available. These intervals are from 02/25/2006




to 03/18/2006 and 03/01/2011 to 06/01/2011. The seismic and discharge data for these intervals were processed identically to the 2017 data. The Northern California Earthquake Data Center is the source of the seismic data for this study and instrument response was causally removed (Haney et al., 2012). The California Department of Water Resources' California Data Exchange Center is the source of all discharge data reported in this study (California Department of Water Resources, 2017c).

## 3.2 Frequency dependent polarization analysis

We expect that contributions to spillway-generated seismic energy will produce energy across a range of frequencies, analogous to observations in natural channels (Gimbert et al., 2014). Energy sources in different frequency bands may also exhibit different particle motion and amplitude. We extract particle motion polarization attributes at each frequency by applying Frequency Dependent Polarization Analysis (FDPA) to the single-station three-component data (Park et al., 1987). The approach in this study is similar to ambient noise analysis applied to seismometer networks (e.g. McNamara and Buland, 2003; Koper and Hawley 2010; Koper and Burlacu, 2015). Following Koper and Hawley (2010), for each component ($u_x$, $u_y$, $u_z$), an hour of record (as ground velocity) is selected and divided into 19 sub-windows that overlap 50%. Each sub-window is tapered with a Hanning window, converted to ground acceleration, and the Fourier transform is computed. At each frequency considered (up to the Nyquist frequency), the Fourier coefficients from each of three components are arranged into a 3x19 matrix, from which the 3x3 cross-spectral covariance matrix is estimated. The eigenvector corresponding to the largest eigenvalue of each 3x3 matrix describes the particle motion ellipsoid within the hour of observation at each frequency (Park et al., 1987). The time averaging inherent to this methodology minimizes the influence of transient seismic sources such as earthquakes or intermittent anthropogenic noise. The application of FDPA is useful for identifying polarization characteristics at a range of frequencies, yet for weakly polarized seismic energy the polarization attributes are highly variable with time. Therefore, it is more meaningful to analyze the probability distributions of polarization attributes in time intervals of similar seismic polarization (Koper and Hawley, 2010).



Earth **Surface**
**Dynamics**
Discussions



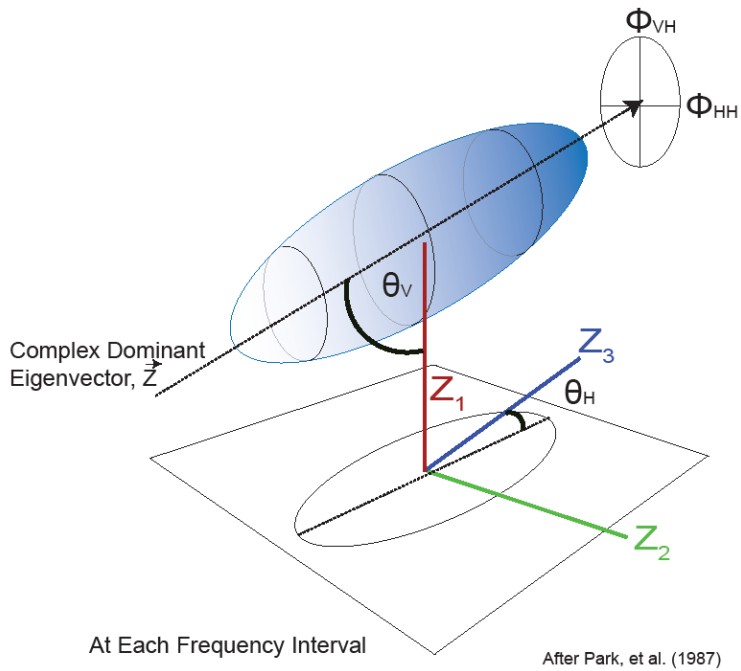

**Figure 3: Diagram of particle motion defined by the dominant eigenvector. The particle motion at each frequency is analyzed by considering the dominant eigenvector of the spectral covariance matrix; the complex-valued components of this eigenvector can be visualized as describing a particle motion in an ellipsoid (Park, et al.,1987). The orientation of the eigenvector and the phase**
5 **relationships between the components of the eigenvector yield the polarization attributes.**

We compute the polarization attributes used in this paper from the complex components of the dominant eigenvector, $\vec{Z}$ [$z_1$, $z_2$, $z_3$] (Fig. 3). For the benefit of the reader, we briefly summarize their computation below and refer the reader to Park et al. (1987) for additional discussion. The azimuth ($\Theta_H$) of the ellipsoid, measured clockwise-from-north, is computed from:

$$\Theta_H = \tan^{-1}\left|\frac{Re(z_3 e^{i\theta_h})}{Re(z_2 e^{i\theta_h})}\right|, \tag{1}$$

where $\theta_h$ is the phase angle:

$$\theta_h = -\frac{1}{2}\arg(z_2^2 + z_3^2) + \frac{l\pi}{2} \tag{2}$$

and $l$ is the lowest non-negative integer value that maximizes the expression:

$$|z_2|^2 cos^2(\theta_h + \arg(z_2)) + |z_3|^2 cos^2(\theta_h + \arg(z_3)) \tag{3}$$

The range of $\Theta_H$ is restricted such that $0° < \Theta_H \le 180°$ if $Re(z_1 z_3^*) < 0$ and $180° < \Theta_H \le 360°$ if $Re(z_1 z_3^*) \ge 0$.

Analogously, the angle of incidence ($\Theta_V$), measured from the vertical, is computed from the major axis of the particle motion ellipsoid as:





$$\Theta_V = \tan^{-1}\left|\frac{Re(z_1 e^{i\theta_v})}{Re(z_H e^{i\theta_v})}\right|, \tag{4}$$

where:

$$\theta_v = -\frac{1}{2}\arg(z_1{}^2 + z_2{}^2 + z_3{}^2) + \frac{m\pi}{2} \tag{5}$$

and $m$ is the lowest non-negative integer that maximizes the expression:

$$|z_1|^2 \cos^2(\theta_v + \arg(z_1)) + |z_2|^2 \cos^2(\theta_v + \arg(z_2)) + |z_3|^2 \cos^2(\theta_v + \arg(z_3)) \tag{6}$$

If $Im\left(\sqrt{z_2{}^2 + z_3{}^2}\right) < 0°$, the sign is reversed to restrict $\Theta_V$ such that $0° < \Theta_V \leq 90°$.

We consider two additional angles to describe the particle motion. First, the phase angle difference between the two horizontal components $z_2$ and $z_3$ ($\phi_{hh}$) of the primary eigenvector, restricted to within -180° and 180°; and second, the vertical-horizontal phase angle difference ($\phi_{vh}$), computed from the phase angle difference between $\theta_h$ (Eq. 2) and $z_1$, restricted to lie between -90° and 90°. Following Koper and Hawley (2010), we also compute the degree of polarization ($\beta^2$) defined by Samson (1983), which is zero when the three component eigenvalues are equal, and is one when the data are described by a single non-zero eigenvalue, such as for a single propagating seismic wave. We emphasize that FDPA methods characterize the dominant seismic source rather than describing the distribution of all sources.

## 4 Results

In the following analysis, we present the polarization attributes in one hour intervals aligned with the hourly discharge data and assume each hour has a consistent seismic character. We then evaluate the variability of all of the hourly polarization attributes within each constant discharge time interval and throughout the dam erosion crisis.

### 4.1 Seismic power variation with changing spillway discharge

We expect the seismic power generated by the flood control spillway to vary with spillway discharge. The power associated with the principal eigenvector of the FDPA polarization ellipsoid during the five constant-discharge time intervals is shown in fig 4. In the figure, the mean hourly power values within each time interval are plotted with a one-standard-deviation envelope representing the variability in power within each constant-discharge interval. In all five time intervals of interest, a microseismic peak between 0.1 and 0.3 Hz is visible, consistent with the ocean-generated microseism (McNamara and Buland, 2004). Interestingly, there is greater "Pre-Chasm" power at frequencies below 0.05 Hz and around 0.25 Hz than the three time intervals after the chasm has developed. This may be attributable to variability in wave heights in the northern Pacific Ocean. The greatest difference between the "Zero Discharge" and all other time intervals is in the 0.5-5 Hz frequency range, with differences of up to ~30 dB between the "Zero Discharge" and "High Discharge" intervals. Spillway turbulence is



therefore observable in this frequency band, even before the beginning of the erosion crisis. Between 0.5 and 1 Hz, the difference in power between the approximately equal discharge "Pre-Chasm" and "Post-Chasm" time interval is greatest, suggesting that increased turbulence resulting from the spillway damage is observable in this frequency band. In the rest of this study, we focus on this frequency range (0.5 to 1 Hz) to evaluate scaling in seismic power and discharge, though differences

in the signal are visible across a broad frequency band (0.2 and 12 Hz). At 0.7 Hz, a peak is prominent in the "Post-Chasm" power, possibly reflecting that the "Post-Chasm" time interval has the most complex channel geometry. These observations indicate seismic power during the five constant-discharge time intervals is sensitive to the turbulent intensity, as inferred from channel geometry.

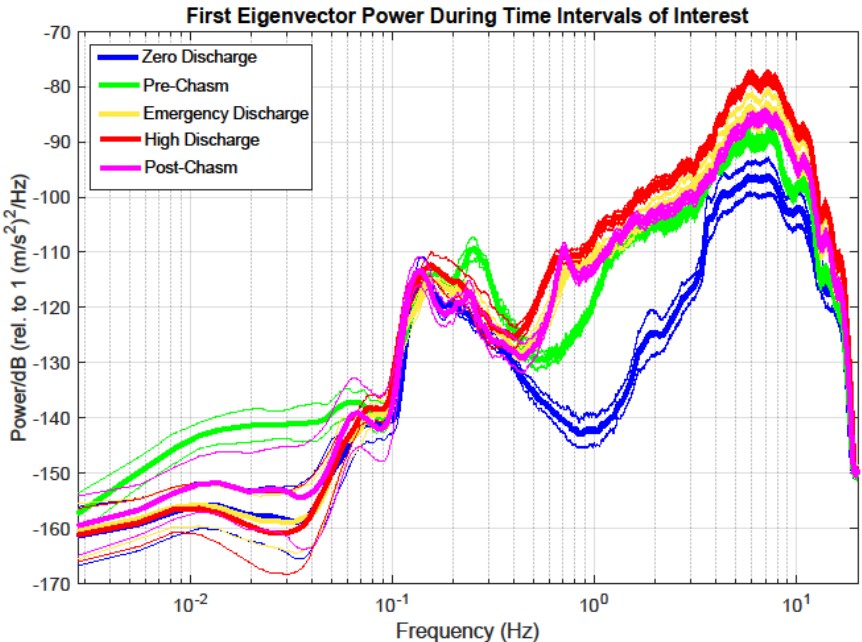

**Figure 4: Power-per-frequency output for each of the five studied intervals, shown with one standard deviation error bars. There is a significant increase (up to 30dB) in the average power of this eigenvector during the four time intervals with discharge, particularly between 0.5 and 12 Hz. The power during three time intervals following spillway damage exceeds the 'Pre-Chasm' at frequencies above 0.5 Hz.**

To further investigate the relationship between seismic power and variations in spillway discharge, we compute the

hourly mean amplitude in the 0.5 to 1 Hz frequency band and compare it to discharge. In fig. 5, the hourly mean amplitude of the first eigenvector is shown for the 2017 crisis period (Fig. 5a) and the 2006 and 2011 release periods (Fig. 5b and Fig. 5c). Figure 5d shows the release discharges of the 2017, 2006, and 2011 releases. Counterclockwise hysteresis is present in the 2017 period containing the erosion crisis, which is not present in 2006 or 2011 periods which maintain a consistent channel form.



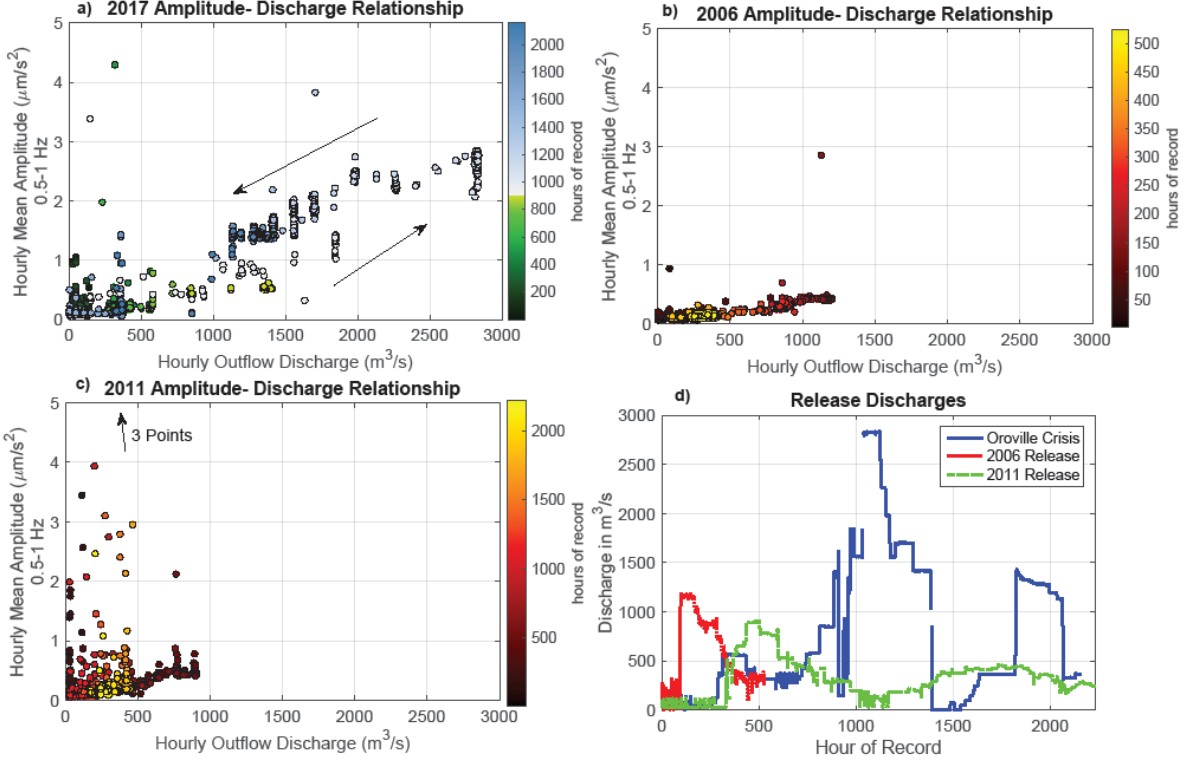

**Figure 5: The plot of mean hourly amplitude of the dominant eigenvector in the 0.5-1 Hz frequency band vs hourly discharge shows that the two correlate strongly. The abrupt change in the colorbar coincides with the timing of the Oroville Dam crisis, and allows two distinct regimes to be identified. Seismic amplitudes are greater by ~0.5 μm s⁻¹ after the uncontrolled channel erosion begins on February 7ᵗʰ, and remains greater even as discharge decreases to earlier levels, demonstrating that hysteresis is observed. This hysteresis is greatest in the 0.5-1 Hz frequency band.**

In figure 6c, the hourly mean power of the first eigenvector is shown for the entire 2017 interval of record as a function of discharge. There is significant variability in hourly mean power for intervals with low discharge, possibly related to other sources of noise including anthropogenic noise created during spillway repair efforts, wind noise, or distant fluvial or marine sources. The scaling of first eigenvector power appears to have a break in slope for low discharge at approximately 200 m³ s⁻¹, which we interpret as the threshold discharge above which signals emanating from the Oroville spillway become the dominant source of seismic energy (i.e. the principal eigenvector of the particle motion ellipsoid in the FDPA). We limit our analysis of scaling between discharge and mean hourly eigenvector power to hours when discharge exceeded 200 m³ s⁻¹, and to hours with spillway use as reported by the California Department of Water Resources. In figure 6a, the scaling relationship between discharge (Q) and power before the crisis is: $P_w \propto Q^{1.75}$. After the spillway defect occurs, the scaling exponent is greater, with $P_w \propto Q^{3.26}$. Figures 6b and 6c display the power-discharge relationships for the 2006 and 2011 release periods.



The scaling exponent for these release events is similar ($P_w \propto Q^{1.69-1.88}$) to the pre-crisis scaling, though there is more scatter in the 2011 seismic record. The change in the scaling relationship between discharge and seismic power is consistent with the inferred change in turbulent intensity following the damage to the flood control spillway.

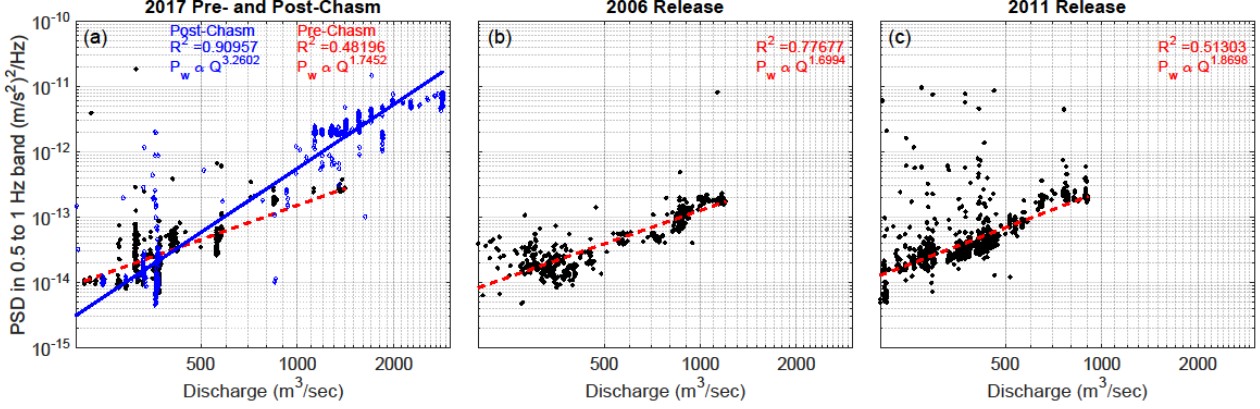

**Figure 6:** Analysis of the relationship between mean dominant eigenvector power and discharge for the current analysis and two previous flood control release events is shown in 6a-6c. The discharge of each interval is shown in Figure 5d. The scaling exponent of seismic power with discharge before the flood control spillway erosion, $Q^{1.75}$, is more similar to the scaling observed with two prior release events with $Q^{1.70}$ and $Q^{1.87}$ in 2006 and 2011, respectively, as compared to a power scaling of $Q^{3.26}$ following the
development of the chasm from erosion.

### 4.2 Polarization attributes

To examine the potential source of seismic waves across a range of frequencies, we display the azimuth and vertical-horizontal phase difference in fig. 7 for the five time intervals of interest. All five polarization attributes are provided in the supplemental materials. To evaluate the variability of polarization within each constant discharge interval, the probability
density functions (PDFs) of all the hourly polarization results are plotted together in fig. 7. In the figure, the polarization attributes are binned into 100 evenly-spaced frequency bins from 0.1 to 15 Hz and the PDFs are normalized so that within in each frequency bin, the probability sums to one. The brighter colors indicate highly focused attributes and the darker colors indicate broadly distributed attributes. When ground motion is insufficiently polarized, polarization attributes are not interpretable (Samson, 1983). We select a cutoff $\beta^2$ at 0.5 as our threshold criterion for interpreting polarization attributes;
Koper and Hawley (2010) selected a $\beta^2$ cutoff value of 0.6. Frequency ranges that are not interpretable by this criterion are shaded grey in figure 7.

The three time intervals after the spillway damage occurred ('Emergency Discharge', 'High Discharge', and 'Post-Chasm Discharge') display similar polarization attributes. The discharge through the emergency spillway weir, which reached a maximum of 360 $m^3$ $s^{-1}$, is masked by the ~1500 $m^3$ $s^{-1}$ discharge in the primary spillway during this time (California
Department of Water Resources, 2017d). When compared to the time intervals with discharge, the 'Zero Discharge' time




interval contains less polarized three-component motion. Based on our threshold criterion, polarization attributes are not interpretable for a broad range of frequencies. At zero discharge, only polarization attributes at frequencies near 1 Hz, 4 Hz, and 10 Hz are interpretable, representing the ambient noise environment of the station. During the four intervals with non-zero discharge, a broad range of frequencies below 12 Hz are interpretable. There is a significant increase in polarization after the

flood control spillway damage in a narrow frequency band around 0.7 Hz. From 1 to 5 Hz, the $\beta^2$ decreases from the "Pre-Chasm" discharge to the three "Post-Chasm" discharge intervals. The decrease in $\beta^2$ may be attributable to a mixing of seismic sources contributing to the ground motion (see discussion).

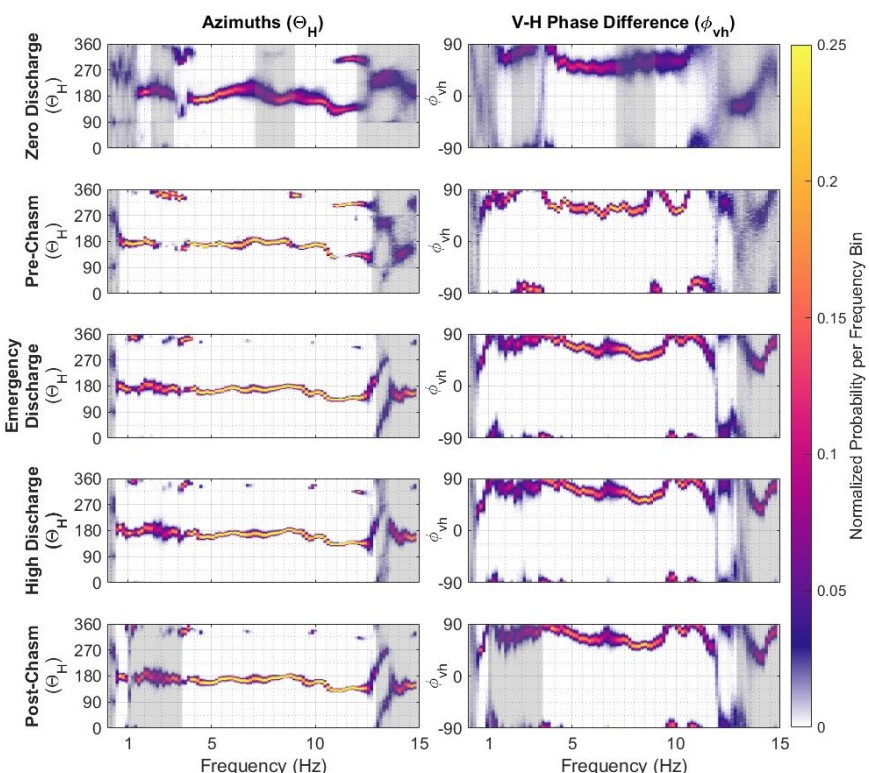

**Figure 7: Two polarization attributes for the five time intervals of interest are presented in two dimensional histograms. Each**
**hour within the time interval of interest has a polarization value at 7201 frequencies. These are distributed among 100 bins evenly spaced in frequency, and are shaded by normalized probability. The polarization attributes for the three intervals of interest after the spillway damage are similar, and differ dramatically from the attributes in the pre-crisis interval. Polarization attributes are interpretable only when the degree of polarization is sufficiently great ($\beta^2$>0.5). Regions shaded grey indicate frequencies at which $\beta^2$<0.5 and the values are not interpretable.**

**4.3 Horizontal azimuth**

To resolve the potential changes in seismic source location resulting from the flood control spillway damage, we evaluate the horizontal azimuth, which is computed for each frequency bin in fig. 7. The horizontal azimuth ($\Theta_H$) of the first eigenvector of the particle motion ellipsoid represents the azimuth of the incoming wave if the motion is Rayleigh-like or a P-wave. Park et al. (1987) and Koper and Hawley (2010) caution interpreting $\Theta_H$ as the azimuth if the horizontal-horizontal



($\phi_{hh}$) phase difference is within 20° of ±90°, because the incidence of a horizontal circular motion is not defined. At zero discharge, the horizontal azimuth is highly variable; multiple sources of seismic energy with equal amplitudes may be present in the absence of spillway discharge (Fig. 7). During the time intervals with spillway discharge, horizontal azimuth is generally consistent from 5-8 Hz, then it stair-steps to lower azimuths at frequencies near 10 Hz.

In order to compute summary statistics of the horizontal azimuth, we select a frequency band of 5-10 Hz. This band has a degree of polarization above 0.5 for all time intervals with discharge and has a horizontal phase angle difference ($\phi_{hh}$) outside of 20° from 90°/-90 (for which the azimuth is not defined). As directional data such as azimuth require special statistical treatment, we employ the CircStat Matlab toolbox for circular statistics to compute an hourly mean azimuth with 95% confidence intervals (Berens, 2009). Due to the 180° ambiguity in azimuth estimates, we consider valid any mean

azimuths that lie between 90° and 270°, and add or subtract 180° from the mean azimuths that lie outside these bounds. This choice is supported by the strong relationship observed between power and changes in discharge which indicate that the flood control spillway channel (between 152° and 183°) is the primary seismic source across a broad range of frequencies (See Fig. 1c). We compute the uncertainty on the mean using 2000 random bootstrap samples with replacement. Table 1 displays the mean 5-10 Hz azimuth within each time interval, with 95% confidence interval error bars. Figure 8a displays the average

hourly 5-10 Hz $\Theta_H$ as a function of flood control spillway discharge, with hourly 95% confidence intervals for the 2017 period. For comparison, fig. 8b and 8c display the same data for the 2006 and 2011 release periods.

| Time Interval | 5-10 Hz Mean $\Theta_H$ (deg.) | Lower 95% CI (deg.) | Upper 95% CI (deg.) |
|---|---|---|---|
| Zero Discharge | 186.76 | 186.67 | 186.87 |
| Pre-Chasm | 174.28 | 174.16 | 174.38 |
| Emergency Discharge | 169.11 | 169.05 | 169.17 |
| High Discharge | 169.78 | 169.73 | 169.82 |
| Post-Chasm | 168.96 | 168.92 | 169.00 |

**Table 1: Distribution statistics for the mean azimuth within the five time intervals of interest. The 95% confidence intervals (CI) on**
**the mean are determined by collecting 2000 random bootstrap samples with replacement.**

At low spillway discharges, the horizontal azimuth values are variable but generally point southward towards the Feather River and town of Oroville (183° to 250°), whereas during time intervals with elevated discharge the azimuth values point more consistently toward the flood control spillway channel, centered at 171°. During times when the spillway is undamaged, the hourly mean azimuth is sensitive to spillway discharge above about 500 m s⁻¹. The hourly mean azimuth

moves from the base of the flood control spillway towards the middle of the spillway with increasing discharge. After the erosion damage begins (Fig. 8a), the azimuths point more towards the top of the chasm, where a large waterfall develops as a




result of the erosion damage. Above 1000 m s$^{-1}$, the azimuths point consistently to the middle of the outflow channel. The azimuths around a discharge of 1400 m$^3$/s are different before the erosion crisis occurred (bright green shading) and after a chasm is present (dark blue shading). This distinction indicates that the FDPA-derived azimuths are sensitive to changes in the turbulence regime under normal spillway operation and when erosion damage is present (see discussion in section 5).

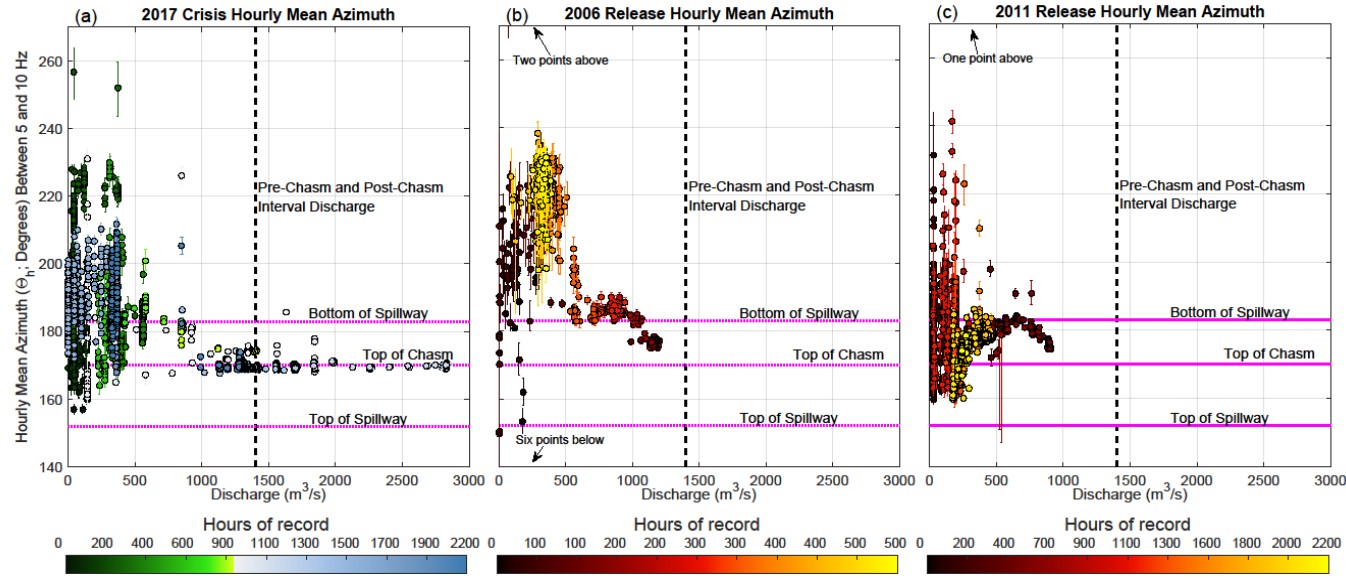

**Figure 8: In the 5-10 Hz band, hourly mean azimuth ( $\Theta_H$ ) is displayed in Fig. 8a-c, with 95% error bars. The mean azimuth is highly variable for discharge less than 500 m$^3$ s$^{-1}$ for the flood control releases in 2017 (Fig. 8a), 2006 (Fig. 8b) and 2011 (Fig. 8c). In Figure 8a, during the "Pre-Chasm" time interval shaded green, the mean horizontal azimuth values point to the bottom of the flood control spillway (183°, see Figure 1c). After the high releases have formed a chasm that starts in the middle of the flood control spillway, the azimuths consistently point to the channel midpoint. The "Post-Chasm" azimuth when discharge is approximately 1400 m$^3$ s$^{-1}$ is noticeably distinct from the "Pre-Chasm" flows around 1400 m$^3$ s$^{-1}$. During times when the channel is undamaged (Fig. 8b and Fig. 8c), the mean azimuth is sensitive to changes in discharge as turbulence develops in the middle of the flood control spillway. Due to the 180° indeterminacy, $\Theta_H$ shown in this figure is constrained between 90° and 270°, the direction of the outflow channel.**

## 4.4 Incident angle

The vertical angle of the first eigenvector of the particle motion ellipsoid represents the incidence angle of the incoming wave for body waves or tilt of elliptical motion for Rayleigh waves. Park et al. (1987) and Koper and Hawley (2010) caution interpreting this metric if $\phi_{vh}$ is within 20° of ±90°, because the vertical incidence angle of vertical circular motion is not defined. At a broad range of frequencies this criterion is not met during time intervals with discharge (see section 4.5). In all five time intervals of interest, the $\Theta_V$ values are highly variable (see supplemental material).



### 4.5 Vertical-horizontal phase difference

To evaluate the possible surface wave type (i.e. Rayleigh or Love), we rely on the vertical-horizontal phase difference. For a Rayleigh wave in an isotropic medium, the vertical-horizontal phase difference will be ±90°. In certain anisotropic structures, the vertical-horizontal phase difference for a Rayleigh wave will deviate from ±90° (Crampin, 1975). In fig. 7, the
vertical-horizontal phase angle ($\phi_{vh}$) is consistently near ±90° for frequencies below 5 Hz when discharge is occurring, which is consistent with a Rayleigh-like wave. At frequencies of up to 8 Hz, which account for most of the power, there is a declining vertical-horizontal phase angle to approximately 50°. At 8 Hz, the vertical-horizontal phase angle is 50° in the "Pre-Chasm" time interval and near 90° in the "Post-Chasm" time interval. These deviations from ±90° are unexpected and explored in Sect. 4.7.

### 4.6 Horizontal phase difference

For all of the time intervals of interest, the $\phi_{hh}$ is between ±180° and ±90° for most frequencies, suggesting horizontal elliptical particle motion. At 8 Hz, the "Pre-Chasm" and "Post-Chasm" time intervals seem to change from near -180° to near -115° phase difference, suggesting a change from linear horizontal motion to more elliptical horizontal motion at frequencies near 8 Hz.

### 4.7 Topographic effects on vertical-horizontal phase angle

To investigate the possible reasons behind deviations from the expected vertical-horizontal phase difference of ±90°, we consider the effect of the hillside topography on the polarization results. To evaluate the influence of local topography on the polarization results, we computed 2D synthetic seismograms using the 2D spectral-element solver package SPECFEM2D 7.0.0 (Tromp et al., 2008; Komatitsch et al., 2012). All geospatial data were processed in ESRI ArcMap 10.4. First, a 2013 ⅓
arc-second resolution digital elevation model was acquired from the USGS National Elevation Dataset at www.nationalmap.gov. The raster was reprojected to Universal Transverse Mercator Zone 10N to acquire northing and easting coordinates in a conformal (angle-preserving) coordinate system. Elevation data (in m, NAVD 88) were extracted from each grid cell along a profile line between top of the spillway erosion damage and the seismometer in this study. The topographic profile was meshed into the model domain using the built-in xmeshfem2d program. To minimize model boundary effects, the
lower model boundary extends over 1000 m below the surface. We also generated a rectangular model grid with a flat surface in SPECFEM2D for comparison. We select a density of 2700 kg m$^{-3}$, P wave velocity of 3000 m s$^{-1}$, and assume a Poisson solid.

In both the topographic and flat surface simulations, a continuous signal was used as the seismic source, and was applied at the location of the midpoint of the Oroville flood control spillway channel. A six-minute random signal varying
between 0 and 1 was filtered using a second order Butterworth filter between 0.5 and 5 Hz, representing the frequencies across which the greatest increases at the BK ORV seismometer with discharge were found (fig. 4). The angle of incidence of the



continuous seismic source was varied between 0°, 45°, and 90° with respect to the vertical. Synthetic 2D seismograms were simulated at the location of the BK ORV seismometer. As this simulation is 2D, the results may only be used to evaluate the effect of topography on vertical-horizontal phase difference. The results of this simulation show that for a vertically incident fluctuating force applied at the Oroville flood control spillway midpoint, the particle motion is Rayleigh-like (vertical-

5    horizontal phase difference is near ±90°) for a flat surface (**Fig. 9a).** As the fluctuating force is applied at angles of 45° and 90° to the surface, the vertical-horizontal phase becomes less Rayleigh-like. Realistic topography also appears to significantly affect the particle motion, which becomes less Rayleigh-like, as vertical-horizontal phase differences decrease from ±90° to ±45° across a range of frequencies (**Fig. 9c**). . This is consistent with the conversion of Rayleigh energy to body-waves as the seismic waves propagate up-slope (e.g. McLaughlin and Jih, 1986).

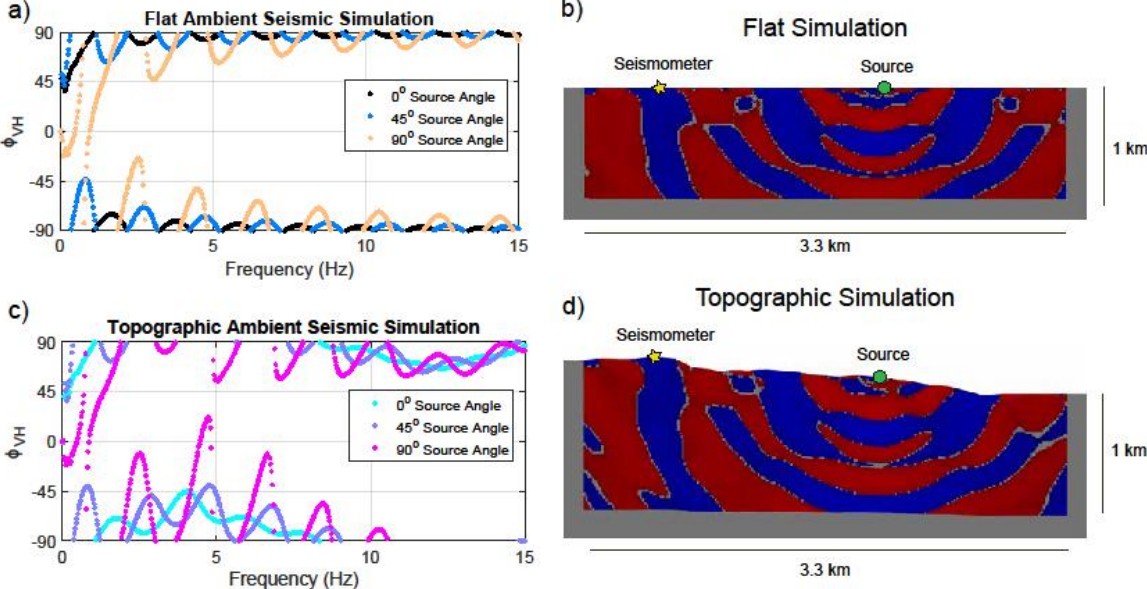

**Figure 9: Polarization attributes computed using FDPA of synthetic seismograms computed using SPECFEM2D are shown in 9a and 9c; with corresponding simulated topographies. Fig 9a and 9b display the horizontal component seismic wavefield during a single time step in each simulation. In the flat surface simulation (Fig. 9a), the vertical-horizontal phase difference is closer to ±90°**

15   **than in the simulation that includes the realistic hillslope topography (Fig. 9c). With a vertically incident force (0° source angle), the phase difference is lowest, while with increasing incidence angles, the vertical motion becomes less like a classical Rayleigh wave.**

## 5 Discussion

The changing geometry of the flood control spillway and the increase in flow turbulence during the Oroville Dam Erosion Crisis are reflected in the FDPA results, most notably in first eigenvector power and horizontal azimuth. During the

20   crisis, large volumes of material (1.3 x 10$^6$ m$^3$ according to our analysis of LiDAR data) were transported, which previous work has shown can contribute to the overall sesimci signal (Tsai, et al., 2012). Therefore, one might expect bedload transport





to be the dominant source of seismic energy. Yet, there are compelling lines of evidence that suggest that the majority of the signal is flow-generated. First, the fastest rate of material transport on the Oroville flood control spillway was likely during the early part of the crisis timeline. Water entering the flood control spillway is from the surface of the reservoir. Unlike a natural river, it does not carry bedload or coarse suspended sediment, so any transported material must be entrained from the spillway

itself or the adjacent hillside. Early in the Oroville dam crisis, weathered saprolite and concrete blocks were undercut and eroded, while later in the crisis, the water from the spillway flowed over harder volcanic rocks. If the seismic signal was generated by a transient transport pulse, we would expect a rapid jump and decay in the amplitude of the seismic waves coming from the spillway. If greater erosion occurred at the beginning of the crisis and if transported material were the primary source of the seismic energy, we would expect clockwise power-discharge hysteresis in this system. Instead, we observe

counterclockwise hysteresis in this relationship. Although our analysis does not enable us rule out all other seismic sources such as material transport, we think that the changes in FDPA results are consistent with changes in the turbulent flow regime caused by erosional changes in channel geometry.

Counterclockwise hysteresis in the discharge-power relationship is consistent with the increased channel roughness and larger scaling of macroturbulent eddies as induced by the Oroville Dam erosion crisis. Because of the dissimilarity of the

system to a natural channel, we are unable fully to implement theoretical models of fluvial seismic energy generation, but we are able to examine whether the scaling relationships within these models are consistent with our data. The theoretical scaling relationship between water-generated vertical component power ($P_W$) and discharge ($Q$) for water turbulence alone with a simple channel geometry is $P_W \propto Q^{1.25}$(Gimbert et al., 2014; Gimbert et al., 2016). Roth et al. 2017, found a $P_W \propto Q^{1.49-1.93}$ in the 35 - 55 Hz band. In the 0.5 to 1 Hz band for the smooth channel (2006, 2011, and pre-crisis 2017) the observed scaling

of first eigenvector power and turbulence is $P_W \propto Q^{1.69-1.88}$, similar to the scaling observed by Roth et al. 2017. After the spillway erosion crisis, the scaling exponent is much higher ( $P_W \propto Q^{3.28}$). We observe similar scaling relationships for the vertical component power, with 2006, 2011, and pre-crisis 2017 scaling as $P_W \propto Q^{1.74-1.98}$ and post-crisis 2017 scaling as $P_W \propto Q^{3.26}$ . The increased scaling exponent following the crisis likely corresponds to the substantially greater turbulence generated from the rough channel morphology and waterfall.

For a uniform turbulent flow, as expected in the hydraulically smooth, constant-width channel geometry present during the 2005-2006 flood, discharge is log-linearly related to flow depth according to the Law of the Wall. For a hydraulically rough channel geometry with variable width, the flow depth is a function of discharge, channel shape, and flow resistance (Leopold et al., 1960). Therefore, determining the causes of the scaling exponent change is difficult. Erosion during the 2017 event incised a 47-meter-deep, V-shaped channel, which increased flow depths from the same discharge and changed the

distribution of shear stresses applied to the bed. Greater flow depths would also allow for larger eddies to form. Our observations support the use of the exponent in the $P_W \propto Q$ power function to observe changing channel geometries in supply-limited fluvial systems (as in Gimbert et al, 2016), but are unable to ascribe a more detailed source mechanism.

The FDPA polarization attributes reveal the seismic character of open channel turbulent flow, which is distinct from the background seismic character ('Zero Discharge' interval) across a broad range of frequencies (fig. 7; supplemental



material). The three time intervals with discharge following the flood control spillway damage have similar polarization attributes, while the "Pre-Chasm" time interval is identifiable by a higher degree of polarization at frequencies below 3 Hz, and the absence of a 0.7 Hz sharp peak in first eigenvector power (fig. 4) and degree of polarization. The decrease in degree of polarization is consistent with mixed seismic waveforms from multiple sources (Rayleigh, Love, P, and S) being introduced

by the chasm channel complexity and increased turbulent intensity. We are unable to attribute a source to the 0.7 Hz anomaly, but we note that around 0.7 Hz we observe azimuths of about 180º, an incidence angle of about 25° from vertical, a vertical-horizontal phase difference about 45°, and broadly distributed horizontal-horizontal phase difference. The azimuth is consistent with the base of the flood control spillway, though the vertical incidence is steeper than the 13° slope of the hillside.

      The greatest hysteresis in the power and discharge relationship is observed at low frequencies (0.5 to 1 Hz), however,

the greatest hysteresis in azimuth is observed at higher frequencies (5-10 Hz). This change may be due to the greater sensitivity to source location that is provided by the higher frequencies, which have shorter wavelengths. For a Rayleigh wave traveling through rock at approximately 3 km s$^{-1}$, the wavelength of a 0.5-1 Hz wave is 6 km to 3 km, significantly longer than the 1 km long flood control spillway, meaning that changes in source location along the spillway may not be observable in azimuths computed at low frequencies. However, at 5 to 10 Hz, the wavelength is 0.6 to 0.3 km, which is sufficient to identify distinct

segments of the flood control spillway.

      The hourly 5-10 Hz mean azimuths (fig. 8) are sensitive to changes in discharge even when no damage is present (fig 8b and 8c). Aerial photographs of the spillway at a range of discharges reveal that the location of the transition from smooth to visibly white and aerated turbulent flow in the bottom half the spillway is sensitive to changes in discharge. In the dam engineering literature, the onset of surface turbulence is referred to as the inception point and represents where the turbulent

boundary layer reaches the free surface (Hunt and Kadavy, 2010). The aerated flow region downstream of the inception point indicates an increase in turbulence and energy dissipation. Due to the geometry of the spillway channel with respect to the seismometer, as the inception point moves up the spillway channel it approaches the seismometer. We expect the closest portion of the aerated flow region to be the largest source of seismic energy under undamaged conditions; seismic energy excited further from the seismometer will be subject to more geometrical spreading and attenuation.

The hourly 5-10 Hz mean azimuths are also sensitive to changes throughout the dam erosion crisis. During the 2017 period, the 'Pre-Chasm' and 'Post-Chasm' time intervals have a statistically significant difference in mean azimuth of 5.32°. The 'Emergency Discharge', 'High Discharge', and 'Post-Chasm' time intervals have mean azimuths within a 1° range. To interpret these results, we reviewed available aerial photography throughout the Oroville Crisis and extracted an elevation profile along the length of the flood control spillway using the LiDAR measurements provided by the CADWR. The imagery

review reveals that the top of the erosion damage propagated upstream a distance of approximately 120 meters (approx. 2.8° azimuth) between February 7$^{th}$ and February 27$^{th}$-28$^{th}$ (fig. 10). The upstream end of the erosion damage forms a waterfall. FDPA results from the 'Emergency Discharge', 'High Discharge', and 'Post-Chasm' time intervals are able to identify the waterfall at the top of the erosion damage. The 'Emergency Discharge' time interval has an azimuth within 1° of the



immediately following 'High Discharge' interval, indicating that 360 m$^3$ s$^{-1}$ released through the emergency spillway did not generate sufficient energy to mask the concurrent flood control spillway releases at that time.

The particle motion of seismic waves produced by the Oroville dam spillway is mostly Rayleigh-like, particularly at frequencies below 3 Hz, though we also observe consistent deviation from the expected Rayleigh $\phi_{vh}$ values (-90° and 90°)
at frequencies from 5-10 Hz. This could be explained by the presence of anisotropy (Crampin, 1975) or Love and/or body waves, which induce shifts in $\phi_{vh}$ but our SPECFEM2D modeling indicates that topography likely influences the polarization attributes we observe, noticeably $\phi_{vh}$. Therefore, our analysis is limited to time-varying changes in polarization attributes rather than interpreting the surface and/or body waveforms created by the flood control spillway. We see the greatest difference in $\phi_{vh}$ and $\phi_{hh}$ between the "Pre-Chasm" and "Post-Chasm" time intervals below 3 Hz and in the 9-11 Hz band, potentially
indicating that more Rayleigh energy is produced at these frequencies after the channel geometry becomes more complex.

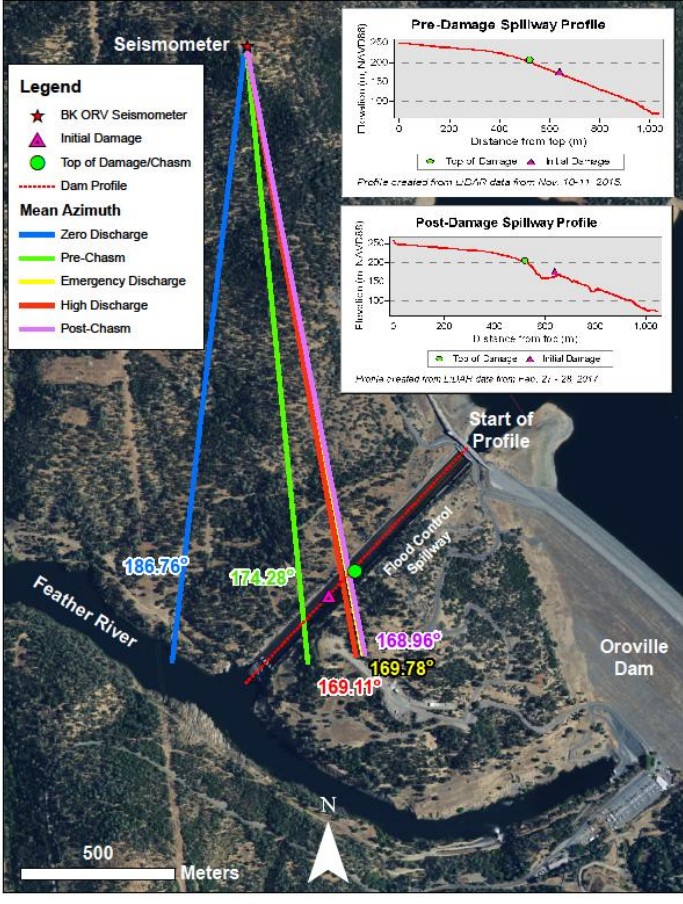

**Figure 10: Mean azimuths for the five time intervals of interest mapped onto aerial imagery reveal the Emergency Discharge, High Discharge, and Post-Chasm mean azimuths point to the top of the spillway damage, where a steep drop creates a waterfall. The location of the initial damage, shown as a triangle, is estimated from photographs of the damage (see supplement). The location of
the damage top, shown as a circle, is estimated from aerial photography and high-resolution LiDAR points collected after most of the damage occurred.**



## 6 Conclusion

Our analysis of the seismic data collected during the Oroville Dam erosion crisis identified several techniques that are potentially useful for dam spillway monitoring and can be applied to fluvial studies. We evaluated the single-station FDPA method to locate the region of greatest flow turbulence. To our knowledge, this is the first application of FDPA methods to analysis of a hydrodynamic signal. We were able to resolve changes in the mean 5-10 Hz azimuth of the turbulence-generated source under normal spillway conditions (2006 and 2011 release periods) when varying discharge and velocity generate changes in the location of the aeration zone inception point. During high spillway discharges and the onset of spillway damage (2017 crisis), the data analysis techniques were used to pinpoint the upstream location of spillway erosion as identified by the increased turbulence. This technique is promising for fluvial studies to identify potential seismic energy interference from nearby waterfalls (i.e. Roth et al. 2016) or in otherwise noisy study environments. The vertical-horizontal phase difference of the spillway-generated energy is consistent with a Rayleigh wave propagating up the dam embankment hillslope.

This study indicated that for constant discharge conditions and varying amounts of spillway damage and associated macroturbulence, counter-clockwise hysteresis in the discharge-seismic power relationship indicates that an increase in turbulence generates more seismic energy. This observation is consistent with the increased energy dissipation by macroturbulent eddies considered in spillway design (Hunt and Kadavy, 2010a). This observation is also consistent with the fluvial geomorphology literature that indicates a significant proportion of total energy dissipation is caused by macroturbulent eddies in natural rivers (Leopold et al., 1960; Bathurst, 1980; Prestegaard, 1983; Powell, 2014). Therefore, seismic monitoring may be a tool to quantify macroturbulent eddies and associated flow resistance in complex natural channels. The results of this study are consistent with those of Roth et al. (2017), who suggested changes in channel morphology as a source for water turbulence-associated hysteresis in natural channels. This study also implies that the Gimbert et al. (2014) model will under-predict seismic energy released in rivers with complex channels and macroturbulent eddies. In this study, we observed that the generation of complex morphology by damage to the spillway produced greater scaling exponents in the seismic power – discharge relationship than the pre-damaged spillway, which produced scaling exponents similar to those predicted by the Gimbert et al. (2014) model.

Although results of this work can be applied to spillway monitoring and natural channel observations, we highlight several limitations of the methods used in this study. The long intervals of constant or known discharge in spillway operations are dissimilar from the sharp increases and decreases in discharge observed in most rivers. In this study, we assumed that during intervals of constant discharge flow turbulence generated seismic motions with the same polarization attributes. Therefore, uncertainty was estimated by documenting the variability of polarization attributes during these time intervals of constant discharge. This study was limited to the hourly resolution of reported discharge and the sampling frequency and sensitivity of the broadband seismometer in the study. For natural rivers, further research is needed to understand the appropriate time window length and sampling frequency to characterize turbulence at various scales.





**Code availability**

The authors provide a MATLAB implementation of the polarization analysis described in the paper, with an example dataset included in the supplemental material.

**Data availability**

The flood control spillway discharge data is available on the California Data Exchange Center (https://cdec.water.ca.gov/). The seismic data used in this study is available through the Northern California Earthquake Data Center (ncedc.org). The LiDAR elevation points and associated metadata provided by the California Department of Water Resources are provided in the supplemental materials.

**Acknowledgements**

This work was supported by the Maryland Water Resources Research Center (project ID: 2017MD341B). The authors thank the California Department of Water Resources for providing the data used in this study. VL acknowledges support from the Packard Foundation. We thank the Computational Infrastructure for Geodynamics (http://geodynamics.org), which is funded by the National Science Foundation under awards EAR-0949446 and EAR-1550901, for access to SPECFEM2D. Data for this study come from the Berkeley Digital Seismic Network (BDSN), doi:10.7932/BDSN, operated by the UC Berkeley

Seismological Laboratory, which is archived at the Northern California Earthquake Data Center (NCEDC), doi: 10.7932/NCEDC. The continuous color scales used in this paper are perceptually uniform and developed by Peter Kovesi and licensed under a Creative Commons license (Kovesi, 2015).

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
