# Peer review of "Seismic signature of turbulence during the 2017 Oroville Dam spillway erosion crisis"

_Earth Surface Dynamics, 2017_

## Referee Comment (RC1) · VC Tsai (Referee) · 30 Jan 2018

VC Tsai (Referee)

tsai@caltech.edu

Received and published: 30 January 2018

In their manuscript, the authors show that the Oroville Dam produced seismic ground motions that can be characterized and which changed with flow characteristics. Overall, the work is reasonably well described, and is a novel application of seismology to observing a process of significant scientific and societal interest. Some of the text and figures can be improved, with more detailed suggestions and comments below, and after such improvements are made, the manuscript should be a nice contribution to the literature.

P1L15,18: See later comments about clarifying discharge scaling and upslope propagation.

P2L17: Run-on sentence.

P2L28: Not clear what is implied by 'geometry variations'.

P3L20: It is not true that the Tsai and Gimbert models assume only Rayleigh waves are excited. In the Tsai model, it is true that a Rayleigh-wave Green's function is used to approximate the response since the force is assumed to be close to vertical, but it is not a limitation of the general modeling framework. In the Gimbert model, a similar approximation is made, but again Love waves could be included in the most generic version of the model.

Figure 1: Panel c needs better labeling. First, it should be clarified where exactly the label 'emergency spillway damage' is referring to. Second, the same names for labels should be used as in panel b. Labels should also be larger, and generally easier to read. Finally, since the distance from the signal to the station is an important parameter, it would be useful to mention somewhere (either in the text or figure) what those distances are. (It can be estimated using the scale bar, but a definite number would be useful.)

P6L1: Listed as 38-hour here but 26-hour in Figure 2. Please clarify.

P6L18: "complied" should be "compiled"

P7L3: "causally" would not be clear to non-seismologists. Either explain in more detail or remove.

Section 3.2 (P8-9): It is not clear that this description is very useful. It is technical, and not that well explained. It might be more straightforward to just describe the statistics used and refer to the references for details, rather than put in a technical section that is challenging to read. Alternatively, the section could be clarified. I believe I understand roughly what the authors did, but this understanding is not from reading the section. As just one example, on P8L7, it is not clear what dominant eigenvector is being discussed. Eigenvector of what?

**ESurfD**
L10P6: "Complex" should be described more.

Figure 5: It is difficult to tell how much of the differences between 2017 and the other years are just due to the difference in range, and how much of the hysteretic behavior is due to something else. In particular, the low-flow part of 2017 does not appear to have strong hysteresis, and is therefore appears quite similar to the other years, and perhaps not distinguishable if the higher flow segments were not there. Incidentally, the color scale chosen for this figure is poor. Please modify to make the times more distinguishable. Potentially larger symbol sizes are needed, or the black edges could be removed to make clearer.

P11L9: Again, first eigenvector of what? Not clear what it is an eigenvector of.

P11L12: Break in slope is not clear. Please clarify.

P12L2-3: This statement needs better explanation. How is the change in scaling relationship consistent with a change in turbulent intensity? Why should the scaling exponent be expected to change in this way, rather than just changing the scale factor, for example (but not the exponent). Somewhere here, it would also be worth commenting on whether the raw signal (without doing a polarization analysis) shows the same behavior or not. Is it necessary to do a polarization analysis? Or is it just clearer using the decomposed polarities? What about the vertical?

Figure 7: Zero discharge azimuths are actually somewhat well determined at a wide range of frequencies. It is true that azimuths are better determined for other times, but only relatively so. So, some discussion should be modified.

P14L24: m/s Units are incorrect

P15L1: m/s Again units are incorrect

P16L26: Do these simulations use uniform velocities? If so, this might yield misleading results, since a more realistic structure in which velocities increase with depth naturally have stronger trapping of waves near the surface, and thus stronger surface waves. (If
simulations use non-uniform velocities, that should be clarified as well.) Partly for this reason, it is not clear how much of this section's analysis really explains the deviations discussed.

P17L9: In a uniform velocity medium with a slope, surface waves simply travel along the slope, rather than horizontally. Part of the complexity shown and cited is due to the non-uniformity of the slope, not just the existence of a slope. This should be clarified.

P18L23: Again, why does greater turbulence imply a change in exponent? This argument needs to be fleshed out, and would add significantly to the conclusions if it can be done quantitatively. It is interesting that the Gimbert model appears to work better during pre-crisis times, but the reason it does not work later should be more specific than the generic 'greater turbulence' statement, since greater turbulence would also just be expected at higher flow rates within the same model.

-Victor Tsai

**ESurfD**

---

## Referee Comment (RC2) · Anonymous Referee #2 · 30 Jan 2018

The manuscript presents seismic analysis of a high discharge event that deeply eroded the flood control spillway at Oroville Dam. Investigation of frequency-dependent 3-component particle motion at the broadband seismometer near the dam allowed continuous estimation of the location of the dominant seismic source and discrimination of the dominant wave type. The circumstance provides an interesting opportunity to investigate the seismic signal of turbulent flow in a channel that initially has a well-known simple shape and lack of bedload. Changes in the seismic signal through the high discharge event are observed and interpreted by the authors in the context of changing turbulent flow conditions in the newly incised spillway channel. Overall the manuscript is well written and effectively presents seismic results relevant to monitoring dams and observing naturally occurring turbulent flows from a safe distance. I think it is suitable

for publication with only minor revisions.

Page 2, Line 28. In this case are the authors referring to changes in channel geometry with time and/or spatially within the channel?

Fig. 1 The bifurcation of the flood control spillway is clear, but the location and type of damage to the emergency spillway is not easy to see. Is the emergency spillway damage meant to refer to the few meters of erosion that appear to be almost uniform along it in the elevation difference map?

Page 11 and Fig. 6. Confidence intervals for the discharge exponent values 'pre' and 'post-chasm' would be useful information. There appears to be a compelling difference, but an attempt to quantify the uncertainty would be an improvement.

Fig. 7. The authors might consider labeling the azimuth corridor that corresponds to the spillway as a handy visual reference. But I understand that it may not be ideal if it obstructs other information.

Section 4.7. This is a good attempt at estimating the effect of topography on the polarization results, and the authors acknowledge some of the limitations of the 2D simulation. I would suggest a bit of additional caution regarding the simple velocity model because the frequency dependent polarization of surface waves could be strongly affected by depth-dependent (and spatially variable) velocity structure likely including anisotropy. I agree that the modeling effort presented provides a viable explanation for some of the deviation from idealized surface wave propagation without topography, I'm just encouraging clear description of its limitations.

Section 4.7. and Fig. 9. Is the oscillating VH angle in Figure 9 because only one point source is considered? Would it be more realistic to sum the seismograms with staggered time shifts to simulate a temporally continuous and spatially distributed source process?

Discussion. The difference in exponent 'pre' and 'post-chasm' is interesting, and even

though there is not a clear explanation for it I think the higher exponent is a useful target for future studies. In regard to comparison with the Gimbert et al. model I wonder if the extreme steepening of the channel to essentially a waterfall into the 'chasm' is beyond the limits of the model formulated by Gimbert et al or if they actually thought the model assumptions would still be reasonably well justified in that setting?

The supplementary material is used appropriately and will be valuable to researchers in the field.

Continuous line numbering would be more helpful for review, but maybe that's a journal policy.

---

## Author Comment (AC1) · 14 Mar 2018

The authors' response, including a revised version of the manuscript, is uploaded in the form of a supplement.

Please also note the supplement to this comment:
https://www.earth-surf-dynam-discuss.net/esurf-2017-71/esurf-2017-71-AC1-supplement.pdf

---

## Author Response (AR1)

**Author's Response To Reviewers**

Title: Seismic signature of turbulence during the 2017 Oroville Dam spillway erosion crisis
Manuscript Number: esurf-2017-71
Paper Submission Date: 09 Dec 2017

We thank our reviewers for their insightful comments. We address each comment below and attach a revised version of the manuscript containing updated figures and with altered text in red. In our comments, the page and line references are to this revised manuscript. The revised version of the paper supplement follows the revised manuscript.

**Reviewer #1 (Victor Tsai) Comments**

1. ***P1L15,18: See later comments about clarifying discharge scaling and upslope propagation.***

Changed "propagation up the hillside topography" to "propagation up the uneven hillside topography"

2. ***P2L17: Run-on sentence.***

Altered phrasing of sentence to:

"Because turbulence affects erosional processes in both hydraulic structures and natural rivers, techniques from the seismic river monitoring (fluvial-seismic) literature provide guidance."

3. ***P2L28: Not clear what is implied by 'geometry variations'.***

Changed "channel geometry variations" to "deviations from spatial uniformity".

4. ***P3L20: It is not true that the Tsai and Gimbert models assume only Rayleigh waves are excited. In the Tsai model, it is true that a Rayleigh-wave Green's function is used to approximate the response since the force is assumed to be close to vertical, but it is not a limitation of the general modeling framework. In the Gimbert model, a similar approximation is made, but again Love waves could be included in the most generic version of the model.***

We appreciate the distinction. The sentence has been changed to:

"While recent forward models to estimate the power spectral density of seismic energy produced by moving bedload and turbulently flowing water can accommodate the excitation of various seismic waves, their applications to date assume that only Rayleigh waves are excited (Tsai et al., 2012; Gimbert et al., 2014). This assumption has not been quantitatively tested."

5. ***Figure 1: Panel c needs better labeling. First, it should be clarified where exactly the label 'emergency spillway damage' is referring to. Second, the same names for labels should be used as in panel b. Labels should also be larger, and generally easier to read. Finally, since the distance from the signal to the***

*station is an important parameter, it would be useful to mention somewhere (either in the text or figure) what those distances are. (It can be estimated using the scale bar, but a definite number would be useful.)*

The figure was updated to have consistent labeling with panel b, the label size was increased, and the outline of the emergency spillway damage was added. The colorscale was also changed to highlight the erosion damage.

On P5L21-22, we added the following line:

"The seismometer is 1.4 km from the top of the flood control spillway channel and 1.9 km from the bottom of the channel".

6. **P6L1: Listed as 38-hour here but 26-hour in Figure 2. Please clarify.**

The time correct length of time is period is 38 hours (between 8am PST on February 11[th] and 10pm PST on Feburary 12[th]). We fixed this in Figure 2.

7. **P6L18: "complied" should be "compiled"**

Fixed.

8. **P7L3: "causally" would not be clear to non-seismologists. Either explain in more detail or remove.**

While that is a technical word, it also has a precise meaning to seismologists and we do not believe that it is too distracting or misleading to the non-seismologists. Therefore, because it adds specificity, we opt to keep it in the sentence.

9. **Section 3.2 (P8-9): It is not clear that this description is very useful. It is technical, and not that well explained. It might be more straightforward to just describe the statistics used and refer to the references for details, rather than put in a technical section that is challenging to read. Alternatively, the section could be clarified. I believe I understand roughly what the authors did, but this understanding is not from reading the section. As just one example, on P8L7, it is not clear what dominant eigenvector is being discussed. Eigenvector of what?**

Thank you for pointing out that this section needs clarification. We provided some elaboration in Section 3.2. We believe the inclusion of this section is justified by the challenging nature of the paper we reference (Park et al., 1987). Our hope is that our description of the polarization attributes will allow the FDPA method to gain wider appreciation and usage.

We have made edits throughout the paper to consistently refer to the eigenvector corresponding to the largest eigenvalue of the spectral covariance matrix at each frequency as the "dominant eigenvector". We now define this usage on P7 L22-25 by inserting:

"Henceforth, we refer to this as the dominant eigenvector. The complex-valued coefficients of this dominant eigenvector describe a particle motion ellipsoid at each frequency, whose properties are analyzed in this paper."

**10. L10P6: "Complex" should be described more.**

Thank you for pointing out that complex is a vague word. We now refer to this as "eroded and incised channel shape."

**11. Figure 5: It is difficult to tell how much of the differences between 2017 and the other years are just due to the difference in range, and how much of the hysteretic behavior is due to something else. In particular, the low-flow part of 2017 does not appear to have strong hysteresis, and is therefore appears quite similar to the other years, and perhaps not distinguishable if the higher flow segments were not there. Incidentally, the color scale chosen for this figure is poor. Please modify to make the times more distinguishable. Potentially larger symbol sizes are needed, or the black edges could be removed to make clearer.**

We have changed the x axis scale in the to the 2006 and 2011 time periods in Figure 5 to make the behavior more visible and have modified the color map, removed the black symbol edge lines, and improved the figure resolution. We agree that no hysteresis is observed in the 2006 and 2011 releases or in the early part of 2017, as stated on P11L4-6. The two releases shown are the largest two available for comparison. For consistency, we changed the color scale, line widths, symbol edge lines, and resolution in Figure 8 as well.

**12. P11L9: Again, first eigenvector of what? Not clear what it is an eigenvector of.**

We have made sure to consistently refer to the eigenvector corresponding to the largest eigenvalue of the spectral covariance matrix at each frequency as the "dominant eigenvector". We now define this usage on P7 L22-25 by inserting:

"Henceforth, we refer to this as the dominant eigenvector. The complex-valued coefficients of this dominant eigenvector describe a particle motion ellipsoid at each frequency, whose properties are analyzed in this paper."

**13. P11L12: Break in slope is not clear. Please clarify.**

By "break in slope", we meant that below the threshold discharge of 200 cubic meters per second, the seismic power does not increase with discharge. We added a figure to the supplement showing the described change in the scaling relationship between dominant eigenvector power and discharge and inserted the following clarification on P12L5-9:

"Below a discharge of approximately 200 $m^3$ $s^{-1}$, there does not appear to be a relationship between dominant eigenvector power and discharge. We therefore interpret 200 $m^3$ $s^{-1}$, as the threshold discharge above which signals emanating from the Oroville spillway become the dominant source of seismic energy. Figure S2 in the supplemental materials shows the dominant eigenvector power for all discharges."

Author's Response To Reviewers

*14. P12L2-3: This statement needs better explanation. How is the change in scaling relationship consistent with a change in turbulent intensity? Why should the scaling exponent be expected to change in this way, rather than just changing the scale factor, for example (but not the exponent). Somewhere here, it would also be worth commenting on whether the raw signal (without doing a polarization analysis) shows the same behavior or not. Is it necessary to do a polarization analysis? Or is it just clearer using the decomposed polarities? What about the vertical?*

On Page 12 we report our observations and develop the possible explanations for these observations in the discussion section. We changed the P12L16-17 to:

"The change in the scaling relationship between discharge and seismic power is consistent with the inferred increase in turbulent energy dissipation following the damage to the flood control spillway (see discussion)".

We included the scale factor (coefficient) in Figure 6 and created a new table (Table 1) to show that both a change in coefficient and exponent are observed as a result of the damage. However, we focus in this paper on the exponent in the power function relating seismic power and discharge rather than the coefficient for the following reasons:

a) The coefficient of the power function reflects the expected seismic power at a discharge of $1 \text{ m}^3 \text{ s}^{-1}$. Our data shows that we are likely not able to observe the spillway-excited seismic energy below a discharge of $200 \text{ m}^3 \text{ s}^{-1}$, and we only use the discharge data above this discharge to fit the power function. We expect the power function coefficient would vary between systems of different sizes (i.e. a small stream and a large river). Since we are only looking at one system in this paper, we only consider the variability in the exponent to be physically meaningful.

b) In the fluvial geomorphology literature, the relationships between discharge and flow velocity, width, depth at a single river cross section are commonly described by power functions (Leopold and Maddock, 1953). This approach is called at-a-station hydraulic geometry, and the exponents of the power functions for flow velocity, width, and depth are used to infer the shape of the channel, channel roughness and, by extension, the degree of turbulence and energy expenditure. To evaluate the similarities and contrasts between rivers of different sizes and discharges, only the scaling exponents are compared.

c) Other seismic studies have similarly characterized the relationship between discharge and seismic power, using the power relationships to infer changes in channel geometry and turbulent state at the same site (Gimbert, et al., 2016; Roth et al., 2017). We wish to place our study in the context of these observations by discussing the power function exponents (the power function coefficients were not reported in Gimbert et al., 2016 or Roth et al., 2017 so we have no basis of comparison).

On page 20 in the discussion and in the conclusion, we have revised our discussion of the water turbulence to more explicitly describe what we mean by "greater turbulence".

"The increased scaling exponent following the crisis likely corresponds to the addition of new sources of turbulent energy dissipation generated from the rougher channel morphology associated with exposed bedrock and

waterfall. For a uniform turbulent flow, as expected in the hydraulically smooth, constant-width channel geometry present during the 2005-2006 flood, discharge is log-linearly related to flow depth according to the Law of the Wall and ground motion is generated by fluctuating forces applied by scaled eddies within the flow, analogous to the processes described by Gimbert et al. (2014). After damage is created in the channel, several mechanisms likely increase the energy dissipated by the flow at a given discharge. The first is that the erosion damage introduced a steep vertical drop in the base of the channel, developing a waterfall. A waterfall will violate assumptions in the Gimbert et al. (2014) model formulation and lead to greater water velocities (from free fall) impacting the bed than would be found in a continuous turbulent channel flow. Second, the irregular channel shape resulting from erosion provides obstructions to the flowing water that create local pressure gradients around the obstacles. These pressure gradients cause a deflection in the flow and an increase in the shearing between flows of different velocities, increasing the energy dissipated by the turbulence in the flow. Third, erosion during the 2017 event incised a 47-meter-deep, V-shaped channel, which increased flow depths for the same discharge and changed the distribution of shear stresses applied to the bed. Greater flow depths would also allow for larger eddies to form. Our results suggest that the additional energy dissipated by these forms of turbulence is observed as an increase in the scaling relationship between discharge and seismic power. Our observations support the use of the exponent in the $P_W \propto Q$ power function to observe changing channel geometries in supply-limited fluvial systems (as in Gimbert et al., 2016), but are unable to identify a particular source mechanism."

The original manuscript contained (on Page 20 Line 28-30) a sentence that addresses the vertical component power scaling (without polarization analysis): "We observe similar scaling relationships for the vertical component power, with 2006, 2011, and pre-crisis 2017 scaling as $P_W \propto Q^{1.74-1.98}$ and post-crisis 2017 scaling as $P_W \propto Q^{3.26}$ ." For clarity, we inserted the parenthesis now on (Page 19 Line34 and Page 20 Lines1-2) so the sentence now reads:

"We observe similar scaling relationships for the vertical component power (without polarization analysis), with 2006, 2011, and pre-crisis 2017 scaling as $P_W \propto Q^{1.74-1.98}$ and post-crisis 2017 scaling as $P_W \propto Q^{3.26}$ ."

15. *Figure 7: Zero discharge azimuths are actually somewhat well determined at a wide range of frequencies. It is true that azimuths are better determined for other times, but only relatively so. So, some discussion should be modified.*

Thanks for helping us see the glass half full ☺. We modified P15L3 to be "somewhat variable"

16. *P14L24: m/s Units are incorrect*

Thank you, fixed

17. *P15L1: m/s Again units are incorrect*

Thank you, fixed

18. ***P16L26: Do these simulations use uniform velocities? If so, this might yield misleading results, since a more realistic structure in which velocities increase with depth naturally have stronger trapping of waves near the surface, and thus stronger surface waves. (If simulations use non-uniform velocities, that should be clarified as well.) Partly for this reason, it is not clear how much of this section's analysis really explains the deviations discussed.***

The simulations as shown in the manuscript used a uniform velocity structure, though we recognize the potential for a velocity structure to influence the results. This comment and comments by Reviewer #2 motivated us to conduct a series of new simulations:

- Introduce a linearly-increasing Vp from 4 km/sec to 6 km/sec while maintaining the Poisson solid assumption.
- Increased the depth of the model domain from 1 km to 4 km to minimize potential effect of domain edges.
- Change the dominant frequencies of the source, which were 0.1 to 5 Hz and represented the largest increase in seismic energy from the spillway as shown in Figure 4. The new dominant source frequencies are between 5 to 10 Hz, which represent the greatest deviations from Rayleigh-like particle motions observed in Figure 7.
- We reduce the simulated time from six minutes to four minutes due to increased computation time due to the larger domain.
- We approximate the source of the seismic energy as along the flood control spillway channel. Due to the position of the spillway channel, the distance between the channel and the station has a range of approximately 500 meters. We simulate the distributed source of the seismic energy as five independently fluctuating point sources along 500 meters of the seismic profile. This simplification ignores the different in hillslope topographies encountered by waves travelling from the top of the spillway and the bottom of the spillway. The below figure is an updated version of Figure S3 in the supplemental materials showing the configuration of the spillway.

[Figure]

- We add random noise to the resulting synthetic seismogram to represent the background seismic energy excited by other sources.

The results of the simulations after making these changes are similar to but much more realistic than those from previous simulations. The simulation with a realistic topography shows a deviation from a Rayleigh-like particle motion of ±90° to ±45° between 5 and 10 Hz, which is similar to the results presented in Figure 7. The updated Figure 9 showing the simulation results is below.

[Figure]

Since our simulations do not consider anisotropy or other possible velocity structures, we softened how we discuss the implications of the simulations. In the abstract, we change "…though numerical modelling indicates these deviations _are_ explained by propagation up…" to "…though numerical modelling indicates these deviations _may be_ explained by propagation up...". We also change the P21L32-33 to "…but our SPECFEM2D modeling indicates that realistic topography is also a viable explanation for the polarization attributes we observe, noticeably $\phi_{vh}$.".

19. **P17L9: In a uniform velocity medium with a slope, surface waves simply travel along the slope, rather than horizontally. Part of the complexity shown and cited is due to the non-uniformity of the slope, not just the existence of a slope. This should be clarified.**

We agree, and clarify this point by inserting "irregular hillside topography" on P17L18 and on P18L15 we replace "… seismic waves propagate up-slope." with "… propagate up a non-uniform slope."

20. **P18L23: Again, why does greater turbulence imply a change in exponent? This argument needs to be fleshed out, and would add significantly to the conclusions if it can be done quantitatively. It is interesting that the Gimbert model appears to work better during pre-crisis times, but the reason it does not work later should be more specific than the generic 'greater turbulence' statement, since greater turbulence would also just be expected at higher flow rates within the same model.**

We addressed this in the response to Comment 14 above. One limitation of this study is that we do not have quantitative information on the turbulence within the flood control spillway. To make this clearer, we inserted the following sentence in P23L27-28: "Due to the hazardous conditions surrounding the spillway channel, inferences on the mechanisms and degree of turbulence are limited to interpretations of aerial photography."

**Reviewer #2 (Anonymous) Comments**

1.   *Page 2, Line 28. In this case are the authors referring to changes in channel geometry with time and/or spatially within the channel?*

For clarity, "channel geometry variations" has been changed to "deviations from spatial uniformity" on P2L28.

2.   *Fig. 1 The bifurcation of the flood control spillway is clear, but the location and type of damage to the emergency spillway is not easy to see. Is the emergency spillway damage meant to refer to the few meters of erosion that appear to be almost uniform along it in the elevation difference map?*

The figure was updated to have consistent labeling with panel b, the label size was increased, and the outline of the emergency spillway damage was added. The colorscale was also changed to highlight the erosion damage.

3.   *Page 11 and Fig. 6. Confidence intervals for the discharge exponent values 'pre' and 'post-chasm' would be useful information. There appears to be a compelling difference, but an attempt to quantify the uncertainty would be an improvement.*

We inserted a table of the 95% confidence intervals of the power function coefficients and exponents. The table as inserted is below:

| Time Interval | Logarithm of Coefficient (Base 10) | 95% Confidence Intervals | | Exponent | 95% Confidence Intervals | |
| --- | --- | --- | --- | --- | --- | --- |
| | | Lower Bound | Upper Bound | | Lower Bound | Upper Bound |
| 2017 Pre-Crisis | -18.055 | -18.438 | -17.671 | 1.7452 | 1.6016 | 1.8888 |
| 2017 Post-Crisis | -22.033 | -22.225 | -21.841 | 3.2602 | 3.1965 | 3.3238 |
| 2006 Release | -17.994 | -18.225 | -17.763 | 1.6994 | 1.6157 | 1.783 |
| 2011 Release | -18.207 | -18.448 | -17.967 | 1.8698 | 1.7776 | 1.962 |

**Table 1: Coefficients, exponents, and uncertainty for power functions fit by least-square regression (shown in Fig. 6).**

4.   *Fig. 7. The authors might consider labeling the azimuth corridor that corresponds to the spillway as a handy visual reference. But I understand that it may not be ideal if it obstructs other information.*

We added dashed lines indicating the azimuth corridor and added the following sentence to the figure 7 caption: "Dashed lines in the first column of figures indicates the azimuth range of the spillway relative to the seismometer (See Fig. 1)."

5.   *Section 4.7. This is a good attempt at estimating the effect of topography on the polarization results, and the authors acknowledge some of the limitations of the 2D simulation. I would suggest a bit of additional caution regarding the simple velocity model because the frequency dependent polarization of surface waves could be strongly affected by depth-dependent (and spatially variable) velocity structure likely including anisotropy. I agree that the modeling effort presented provides a viable explanation for some*

> *of the deviation from idealized surface wave propagation without topography, I'm just encouraging clear description of its limitations.*

The simulations as shown in the manuscript used a uniform velocity structure, though we recognize the potential for a velocity structure to influence the results. This comment and comments by Victor Tsai (Reviewer #1) motivated us to change our simulation in the following ways:

- Introduce a linearly-increasing Vp from 4 km/sec to 6 km/sec while maintaining the Poisson solid assumption.
- Increased the depth of the model domain from 1 km to 4 km to minimize potential effect of domain edges.
- Change the dominant frequencies of the source, which were 0.1 to 5 Hz and represented the largest increase in seismic energy from the spillway as shown in Figure 4. The new dominant source frequencies are between 5 to 10 Hz, which represent the greatest deviations from Rayleigh-like particle motions observed in Figure 7.
- We reduce the simulated time from six minutes to four minutes due to increased computation time due to the larger domain.
- We approximate the source of the seismic energy as along the flood control spillway channel. Due to the position of the spillway channel, the distance between the channel and the station has a range of approximately 500 meters. We simulate the distributed source of the seismic energy as five independently fluctuating point sources along 500 meters of the seismic profile. This simplification ignores the different in hillslope topographies encountered by waves travelling from the top of the spillway and the bottom of the spillway. The below figure is an updated version of Figure S3 in the supplemental materials showing the configuration of the spillway.

[Figure]

- We add random noise to the resulting synthetic seismogram to represent the background seismic energy excited by other sources.

The results of the simulations after making these changes are similar to but much more realistic than those from previous simulations. The simulation with a realistic topography shows a deviation from a Rayleigh-like particle motion of ±90° to ±45° between 5 and 10 Hz, which is similar to the results presented in Figure 7. The updated Figure 9 showing the simulation results is below.

[Figure]

Since our simulations do not consider anisotropy or other possible velocity structures, we softened how we discuss the implications of the simulations. In the abstract, we change "…though numerical modelling indicates these deviations *are* explained by propagation up…" to "…though numerical modelling indicates these deviations *may be* explained by propagation up...". We also change the P21L32-33 to "…but our SPECFEM2D modeling indicates that realistic topography is also a viable explanation for the polarization attributes we observe, noticeably $\phi_{vh}$.".

6.  *Section 4.7. and Fig. 9. Is the oscillating VH angle in Figure 9 because only one point source is considered? Would it be more realistic to sum the seismograms with staggered time shifts to simulate a temporally continuous and spatially distributed source process*

Thank you very much for this helpful suggestion! As you suggest, we now approximate a distributed source by five point sources that independently fluctuate with a dominant source frequency of 5-10 Hz. We also add in a low level of random noise to resulting synthetic seismograms to approximate background seismic noise. The oscillating VH angle behavior is diminished from both of these steps, and we believe the VH angle behavior is more realistic.

7.  *Discussion. The difference in exponent 'pre' and 'post-chasm' is interesting, and even though there is not a clear explanation for it I think the higher exponent Is a useful target for future studies. In regard to comparison with the Gimbert et al. model I wonder if the extreme steepening of the channel to essentially a waterfall into the 'chasm' is beyond the limits of the model formulated by Gimbert et al or if they actually thought the model assumptions would still be reasonably well justified in that setting?*

Clarified in the discussion. A waterfall certainly violates the assumption of the model. We make this more explicit in the discussion on P20L10-12 by including the sentence "A waterfall will violate assumptions in the

Gimbert et al. (2014) model formulation and lead to greater water velocities (from free fall) impacting the bed than would be found in a continuous turbulent channel flow."

*The supplementary material is used appropriately and will be valuable to researchers in the field.*

Thank you.

8. *Continuous line numbering would be more helpful for review, but maybe that's a journal policy.*

We agree, but page-based line numbering is derived from the journal template's format.

[revised manuscript text omitted]
 (CADWR) following an information request on June 30th, 2017. The raw data is provided below with CADWR permission granted 10/19/2017.

November 10th and 11th LiDAR Acquisition: The LiDAR survey was accomplished using an Optech Orion M300 LiDAR system operating from a fixed wing aircraft (Cessna 310 Tail # N7516Q). The mission was completed over two days (November 10 and 11, 2015). A Trimble R8-3 GPS receiver was set up and operating at the Oroville Municipal Airport for the duration of the mission, recording data at 2 Hz.

The March 23rd merged LiDAR dataset provided by the California Department of Water Resources consists of the following datasets. From the metadata associated with the files and information provided by the CADWR, the main spillway damage area surveyed on February 27th and 28th.

- Towill, Inc. 2/24/2017 LiDAR (Additional metadata in file included in .zip file folder)

[Figure]

Oroville_Spillway_02
-24-2017_Project_Me

- CADWR 2/27/2017 Drone Point Cloud (Gated Spillway with no water)
- Towill Inc. 2/28/2017 LiDAR (Additional metadata in file included in .zip file folder)

[Figure]

Oroville_Spillway_02
-28-2017_Project_Me

- CADWR 3/13/2017 Drone Point Cloud (DF1223 upper spillway)
- CADWR 3/19/2017 Drone Point Cloud (DF1300 spoils near Hyatt Powerplant)
- CADWR 3/19/2017 Drone Point Cloud (DF1151 spoils on hillside near auxiliary spillway)

The surveys were conducted with horizontal control in California Coordinate System (CCS) State Plan Zone II (US Feet) and vertical control in North American Vertical Datum (NAVD) 1988 (US Feet).

The raw data is available file included in .zip file folder:

[Figure]

20170323_Oroville_LiDAR.TXT

20151111_Oroville_LiDAR.txt

**Seismic signature of turbulence during the 2017 Oroville Dam spillway erosion crisis (Supplemental Materials)**
**Goodling, Prestegaard, and Lekic (2018)**

In this data, the first column is the easting, the second column is the northing, and the third column is vertical elevation (all in US feet). To create a difference map, a triangle irregular network (TIN) was created from each point dataset. Using nearest neighbor interpolation, a 1-meter resolution digital elevation model (DEM) was created. A difference raster dataset was created by subtracting the 2015 DEM from the 2017 DEM, and converting the result to meters. The volume change in the main spillway damage zone is the sum of each cell's volume (cell area x vertical change). All processing was completed in ArcMap 10.4 (ESRI).

**Polarization Attributes**

[Figure]

**Figure S1-** The four polarization attributes and degree of polarization ($\beta^2$) for the five time periods of interest. Grey shading indicates frequencies at which the polarization attributes are not interpretable ($\beta^2 < 0.5$). The Azimuth ($\theta_H$) and vertical-horizontal phase difference ($\phi_{vh}$) are shown in Figure 7 of the main text.

**Scaling of Dominant Eigenvector Power and Discharge**

**Seismic signature of turbulence during the 2017 Oroville Dam spillway erosion crisis (Supplemental Materials)**

**Goodling, Prestegaard, and Lekic (2018)**

[Figure]

**Figure S2-** The hourly relationship between dominant eigenvector power and discharge has an apparent break in slope at approximately 200 cubic meters per second of discharge. We interpret this to be the threshold for which the seismometer is sensitive to flood control spillway discharge, and complete the scaling analysis in the main text only for hours with discharge greater than 200 cubic meters per second.

**SPECFEM2D Simulation**

Topographic model domain for the SPECFEM2D simulation was created by extracting the elevation profile along a transect extending through the BK ORV seismometer and the center of the Oroville Dam Spillway. To create the model domain, 1000 meters were added to the lowest elevation, so that the model boundary did not interfere with the topography.

**Seismic signature of turbulence during the 2017 Oroville Dam spillway erosion crisis (Supplemental Materials)**

**Goodling, Prestegaard, and Lekic (2018)**

[Figure]

**Figure S3-** The hillside topography extracted for the SPECFEM2D simulation. If the entire length of the flood control spillway is considered an ambient seismic source, then the seismic waves travel a range of approximately 500 meters to the seismometer. In our simulation, we simplify this by simulating five sources spaced 100 meter apart along a profile line to the middle of the flood control spillway.

**Seismic signature of turbulence during the 2017 Oroville Dam spillway erosion crisis (Supplemental Materials)**
**Goodling, Prestegaard, and Lekic (2018)**

[Figure]

**Figure S4-** The elevation profile extracted for the SPECFEM2D simulation.

**Seismic signature of turbulence during the 2017 Oroville Dam spillway erosion crisis (Supplemental Materials)**

**Goodling, Prestegaard, and Lekic (2018)**

a)

[Figure]

b)

[Figure]

c)

[Figure]

**Seismic signature of turbulence during the 2017 Oroville Dam spillway erosion crisis (Supplemental Materials)**

**Goodling, Prestegaard, and Lekic (2018)**

g)

[Figure]

h)

[Figure]

**Seismic signature of turbulence during the 2017 Oroville Dam spillway erosion crisis (Supplemental Materials)**

**Goodling, Prestegaard, and Lekic (2018)**

[Figure]

[Figure]

[Figure]

**Figure S5 (a-f)-** Aerial photographs collected during the time period of interest showing the evolution of the spillway erosion damage.

**Seismic signature of turbulence during the 2017 Oroville Dam spillway erosion crisis (Supplemental Materials)**

**Goodling, Prestegaard, and Lekic (2018)**

[Figure]

**Figure S6**- Three other dams- Thermalito Forebay Dam, Thermalito Diversion Pool Dam, and Thermalito Afterbay Dam- are a part of the Oroville Dam Complex and are at backazimuths of 248°, 232.5°, and 227°, respectively. The town of Oroville, California is located between station backazimuths of approximately 248° and 217°.